# The Evolution and Functional Roles of *miR408* and Its Targets in Plants

**DOI:** 10.3390/ijms23010530

**Published:** 2022-01-04

**Authors:** Yu Gao, Baohua Feng, Caixia Gao, Huiquan Zhang, Fengting Wen, Longxing Tao, Guanfu Fu, Jie Xiong

**Affiliations:** 1College of Life Sciences and Medicine, Zhejiang Sci-Tech University, Hangzhou 310018, China; gaoyu960618@163.com (Y.G.); gao15234867901@163.com (C.G.); Zhqyxdz@163.com (H.Z.); wft1538352651@163.com (F.W.); 2National Key Laboratory of Rice Biology, China National Rice Research Institute, Hangzhou 310006, China; fengbaohua@caas.cn (B.F.); lxtao@mail.hz.zj.cn (L.T.); 3Zhejiang Province Key Laboratory of Plant Secondary Metabolism and Regulation, Hangzhou 310018, China

**Keywords:** evolution, *miR408*, plant development, stress response, yield

## Abstract

MicroRNA408 (*miR408*) is an ancient and highly conserved miRNA, which is involved in the regulation of plant growth, development and stress response. However, previous research results on the evolution and functional roles of *miR408* and its targets are relatively scattered, and there is a lack of a systematic comparison and comprehensive summary of the detailed evolutionary pathways and regulatory mechanisms of *miR408* and its targets in plants. Here, we analyzed the evolutionary pathway of *miR408* in plants, and summarized the functions of *miR408* and its targets in regulating plant growth and development and plant responses to various abiotic and biotic stresses. The evolutionary analysis shows that *miR408* is an ancient and highly conserved microRNA, which is widely distributed in different plants. *miR408* regulates the growth and development of different plants by down-regulating its targets, encoding blue copper (Cu) proteins, and by transporting Cu to plastocyanin (PC), which affects photosynthesis and ultimately promotes grain yield. In addition, *miR408* improves tolerance to stress by down-regulating target genes and enhancing cellular antioxidants, thereby increasing the antioxidant capacity of plants. This review expands and promotes an in-depth understanding of the evolutionary and regulatory roles of *miR408* and its targets in plants.

## 1. Introduction

MicroRNAs (miRNAs) are small noncoding RNAs with 20 to 24 nucleotides (nt) that regulate gene expression post-transcriptionally through base pairing with their complementary mRNA targets [1]. MicroRNA (MIR) genes encoding miRNAs are transcribed into primary miRNAs (pri-miRNAs) by RNA polymerase II (Pol II), and then these pri-miRNAs are processed into miRNA/miRNA* duplexes by the RNase III family enzyme DICER-LIKE1 (DCL1). Subsequently, these duplexes are 2’-O-methylated by the methyltransferase HUA ENHANCER1 (HEN1) at the 3’ end. One strand of the duplex is incorporated into ARGONAUTE1 (AGO1) to form an active RNA-induced silencing complex (RISC). MicroRNA/target complementarity guides RISC to target the mRNA slicing [2,3]. Lin-4 was the first miRNA discovered in the nematode *Caenorhabditis elegans* in 1993 [4]. A second miRNA named let-7 was discovered in 2000 [5]. Since then, thousands of miRNAs have been quickly discovered in animals and plants, and their regulatory roles in many biological processes have been covered in [1,6,7,8,9,10]. In plants, miRNAs are observed to be closely related to transcriptional gene silencing (TGS), indicating that miRNAs play significant roles in plants [9].

MicroRNAs are the second most abundant plant sRNA class [11]. Many plant miRNA families have been conserved for a long time in the evolution of land plants [12]. Axtell and Bartel [12] demonstrated that at least eight miRNA families remained basically unchanged before the emergence of seed plants, and at least two families (miRl60 and miR390) have remained unchanged since the last common ancestor of mosses and flowering plants. Based on a combination of experiments and computations, Fattash et al. [13] identified 48 novel miRNAs, including *miR408*, *miR536*, *miR535,* etc., in the moss *Physcomitrella patens*. Furthermore, the results of RNA degradome analysis revealed that *miR408* was conserved between liverworts and other land plants [14]. In addition, *miR408* was highly conserved in *Oryza sativa*, Medicago and Populus [10]. Nowadays, *miR408* has been widely treated as an ancient and conserved miRNA in plants.

Although *miR408* is ancient and conserved in plants, our understanding of the functions of *miR408* and its targets has just begun. Since *miR408* was first discovered in *Arabidopsis thaliana*, the crucial roles of *miR408* in plants have been verified in many studies [10], and *miR408* has been regarded as important regulator of plant vegetative growth and reproductive development. Because it can effectively avoid the artifacts caused by overexpression mutants, loss-of-function mutants are widely used to study the function of miRNAs. A loss-of-function mutant is regarded as an important prerequisite for understanding the biological function of miRNAs [15]. A recent study showed that the function loss of *miR408* negatively regulated light-dependent seed germination [16]. Zou et al. [17] proved that the function loss of *miR408* positively regulated the synthesis of Salvianolic acid B (SalB) and rosmarinic acid (RA) in *Salvia miltiorrhiza*. T-DNA insertion mutants of *miR408* were used to explore how *miR408* influences copper- and iron-dependent metabolism [18]. In addition, the function loss of *miR408* also caused expression disorders of other miRNAs [19]. In contrast, overexpression mutants have been widely used to discover the function of the target genes of *miR408*. Overexpression of *miR408,* by down-regulating the target genes, can obviously promote numerous development processes, including the growth and development of leaves, flowers, roots, and seeds [20,21,22,23,24,25,26]. Similarly, overexpression of *miR408* can also significantly increase photosynthetic activity, biomass and seed yield by down-regulating target genes [20,21,22,23,24,25]. The overexpression of *miR408* also altered heading times in *Triticum aestivum* [24]. In addition, *miR408* responds to various environmental factors, such as cold, salinity, drought, nutritional deprivation, oxidation, osmotic stress, heavy metals, lipopolysaccharide (LPS) and disease stress [22,27,28,29,30,31,32].

Although previous research has shown that miRNAs play multiple important regulatory roles, there are relatively few studies on the evolutionary pathway. In addition, previous studies either only used overexpression mutants or only loss-of-function mutants, lacking a systematic analysis of the functional effects of *miR408* and its target genes.

## 2. Evolution Analysis of the *miR408* Family in Plants

*miR408* is an ancient and conserved miRNA in plants, a thorough and systematic understanding of its evolution process will help us understand and master the evolution and functional roles of *miR408* in plants more comprehensively. Here, we analyzed the evolution of *miR408* using bioinformatics methods.

### 2.1. Origin and Distribution of miR408 Family Members in Plants

As a typical multifunctional miRNA, *miR408* has been annotated in more than 30 different plants [24]. In order to reveal the origin and distribution of *miR408* family members in plants, we downloaded all *miR408* precursor and mature sequences of known plants from the miRbase database (version 22.0) (http://www.mirbase.org/) (accessed on 31 October 2021) (Appendix A), and carried out biological classification. The results are summarized in Appendix A.

The sequence length of *miR408* varies in different plants. As shown in Appendix A, the sequence length of the *miR408* precursor varies greatly, ranging from 71 to 286 nt, while the sequence length of mature *miR408* is between 20 and 22 nt, with little difference. In addition, the length of the precursor and mature *miR408* are different in the same species.

Usually, the length of a miRNA precursor is about 70–80 nt, and it is very likely that the two arms will produce miRNA separately, so -5p/-3p will be added after the name to distinguish them. As shown in Appendix A, the *miR408* family members include 55 precursor sequences and 72 mature sequences. Some *miR408* precursors are produced separately in the two arms. For instance, the *miR408* precursors produce miRNA in the two arms of *A. thaliana*, *O. sativa*, *Populus trichocarpa*, etc., separately. Among the 72 *miR408* mature bodies, 17 were from the 3p arm and 17 were from the 5p arm. These 34 *miR408* mature bodies came from 17 different angiosperms. In other *miR408* mature bodies from bryophytes, ferns and gymnosperms, whether they were from the 3p arm or the 5p arm was not distinguished.

*miR408* is widely distributed in different plants. As shown in Appendix A, *miR408* family members have been identified in 40 different plants, and distributed in 18 orders and 22 families, most of which belong to Rosales and Gramineae. Among these 40 plants, dicots are the most common, and there are only 10 monocots. In addition, different species have different amounts of *miR408*. For instance, *Physcomitrella patens* has two precursors, while *Selaginella moellendorffii* has only one precursor. Furthermore, *A. thaliana* has two mature *miR408*, while *Saccharum officinarum* has five mature *miR408*. In addition, these 40 kinds of plants also include 1 species of bryophyte, 1 species of pteridophyte, 2 species of gymnosperms and 36 species of angiosperms. Interestingly, since bryophytes represent the first green plants to colonize terrestrial plants [33], it is clear that the *miR408* family is an ancient and widely distributed family of miRNAs and bryophytes may be the evolutionary ancestors of the *miR408* family and since then have been strongly conserved.

### 2.2. Evolution Characteristics of the miR408 Family in Plants

In order to reveal the evolutionary characteristics of *miR408* family member in plants, we respectively constructed the phylogenetic trees of the precursor sequences and the mature sequences of the *miR408* family in Figure 1 and Figure 2.

Species specificity is an important factor affecting the evolution of precursors in the *miR408* family. As shown in Figure 1, except for a few species, the precursors of *miR408* family members of most species have a closer aggregation at the family level, such as Gramineae and Leguminosae in monocots and Solanaceae in dicots. However, the precursors of *miR408* family members in Brassicaceae are more dispersed, and they are distributed on different small branches.

Sequence conservation is one of the main factors affecting the evolution of the mature *miR408* family. Pan et al. [20] showed that, except for the majority of the 5′ nucleotides, the mature *miR408* sequence found in moss, angiosperms and gymnosperm species was unchanged. The results in Figure 2 show that the members of the mature sequences of *miR408* are obviously divided into two branches. The sequences of *miR408*-5p members are more concentrated and form a single branch, and the *miR408*-3p members are gathered in another branch. In addition, as shown in Appendix A, most of the *miR408*-3p mature member sequences are the same except for the first and last bases. However, most *miR408*-5p mature member sequences are quite specific, only a few mature *miR408*-5p family members of the same family or species have relatively conserved sequences. Therefore, the members of *miR408-*5p were quite specific, while the members of *miR408*-3p were conservative.

In addition, among 72 mature *miR408*s, 46 are distributed in dicots, 21 are distributed in monocots, 2 are distributed in coniferopsida, 2 are distributed in musci, and only 1 is distributed in lycopodiinae. In summary, ancient low-grade plants contain fewer *mi**R408* species, and higher dicotyledonous plants contain more *mi**R408* species, indicating that *mi**R408* evolved gradually from bryophytes and played important roles in the evolution of plants to dicotyledonous plants. It may have been the emergence and change of *mi**R408* that led to the evolution of various higher plants species. Members of the *miR408* family coexist with conservatism and specificity in evolution, providing a valuable reference for exploring the various biological functions of the *miR408* family in plants.

## 3. Overview of the *miR408*-Regulated Target Genes

With the help of advanced genomic technologies such as next-generation sequencing, more and more *miR408* target genes have been screened, some of which have been identified using the modified 5′ rapid amplification of cDNA ends (5′-RACE) method. Here, we have summarized the types and functions of the target genes regulated by *miR408* in Figure 3. We have also sorted out and summarized the species distribution and splicing sites of these target genes in Appendix A.

### 3.1. Types of the miR408-Regulated Target Genes

Due to the imperfect complementarity required for binding, miRNAs may target hundreds of mRNAs, but only a small part of these interactions have been experimentally verified [34]. Although *miR408* has different target genes in different species, all these target genes encode Cu proteins [25].

At present, the 5′-RACE technique has been used to validate miRNA targets, as cleavage occurs opposite, and at around the tenth position from the 5′-end of the miRNA [35]. Many *miR408* target genes, such as *plantacyanin* (*PLC*), *uclacyanin* (*UCL*), *cupredoxin*, *laccase* (*LAC*), *chemocyanin-like protein gene*, *timing of CAB expression 1* (*TaToC1*), *DOB1/SK12/helY-like DEAD-box Helicase* (*DSHCT*), *3-ketoacyl-CoA synthase 4 gene* (*KCS*) and *galacturonosyltransferase 7-like gene* (*GAUT*) have been characterized in different plants and well verified experimentally using a modified version of the 5′-RACE polymerase chain reaction (PCR) (Appendix A) [20,21,22,24,25,32,36,37,38,39,40,41,42,43,44]. In addition, *miR408* also targets other genes encoding Cu proteins, such as *copper P1B-ATPase* in *M. truncatula* [42].

### 3.2. Functions of miR408-Regulated Target Genes

Many of the target genes regulated by *miR408* belong to the Phytocyanins (PCs). PCs are blue Cu proteins that bind only one Cu ion. It is not only related to the activity of the electron carrier, but also has a great influence on the growth and stress resistance of plants [25]. Previous studies have shown that PCs are involved in a variety of plant activities, including pollen tube germination [45,46], anther pollination [45,46], reproductive potential determination [47] and apical bud organ development [48]. Most PCs are chimeric arabinogalactan proteins (AGPs), which are subdivided into the stellacyanins, PLCs, and UCLs based on their predicted Cu-binding ligands (His, Cys, His and Met/Gln) [49].

*PLC* is the most widely studied *miR408* target gene in plants. It encodes a small extracellular matrix (ECM) protein, which belongs to the ancient plant-specific phytocyanin family [50], and the complementary sites are conserved in *Spinacia oleracea*, *Zea mays*, *Cicer arietinum* and *O. sativa* [21]. The first *PLC* with functional characteristics is Lily chemocyanin, which is secreted from the pistil and acts as an external signal to regulate pollen tube reorientation in vitro [45]. In addition, plant *PLC* is involved in the process of seed germination, electron transfer and shuttling between proteins, directing pollen tube growth, stress response, Cu homeostasis, plant reproduction, intercellular signal transduction, lignin formation, and so on [16,22,45,46,51,52].

UCL is a member of the ancient plant-specific phytocyanin family, which is a subfamily of blue Cu proteins [52,53]. Nersissian et al. [52] reported that UCLs were a mature form of chimeric proteins, consisting of a Cu-binding domain and a domain resembling a cell wall structural protein. Interestingly, recent research showed that UCL8 is localized in the cytoplasm and is associated with photosynthesis and grain yield, but many other phytocyanins are localized in the plasma membrane [25]. Further analysis showed that *OsUCL8* is related to pollen tube growth, pollination and the seed setting rate of *O. sativa* [26]. In addition, *UCL* acts as a shuttle for electron transfer between proteins [52].

*LAC* is another widely studied target of *miR408*, and was first discovered in 1883 [54]. LAC belongs to the blue oxidases and also belongs to the superfamily of multicopper oxidases (MCOs). It contains a group of enzymes including many proteins with different substrate specificities and multiple biological functions [25,54]. LAC combines four Cu ions and is widely distributed in Anacardiaceae and other higher plants, including *Pinus taeda*, *Acer pseudoplatanus*, *Nicotiana tabacum*, *P. trichocarpus*, *Liridendron tulipifera*, *Lolium perenne*, *A. thaliana*, *Z. mays*, *O. sativa*, *Sacchorum officinarum*, *Brassica napus*, and *Brachypodium distachyon* [54]. In addition, it is reported that plant *LACs* are involved in the process of lignin synthesis [55,56], the maintenance of cell wall structure and integrity [53], responses to environmental stresses [57], wound healing [36], iron metabolism [58] and the polymerization of phenolic compounds [54].

In addition to *PLC*, *LAC* and *UCL*, *miR408* also targets many other genes. *Plastocyanin*, *cupredoxin* and *chemocyanin* are considered to be the electron transfer shuttles between proteins [51,59,60]. *TaTOC1s* are members of the phytocyanin family [24]. Zhao et al. [24] reported that *miR408* regulates the heading time of *T. aestivum* by controlling the transcription of *TaTOC1s*. In addition, *TaTOC1s* are involved in the clock’s evening loop [61]. *IbGAUT* plays a role in the cell wall structure for wound healing [62]. A *chemocyanin-like protein gene* (*TaCLP1*), plays positive roles in the response of wheat to high-salinity, heavy Cu stress and stripe rust [32].

### 3.3. Species Distribution and Splicing Sites of miR408-Regulated Target Genes

Plant miRNA-mediated transcript cleavage is usually located at the 10th and 11th nucleotides of the 5′ end of miRNA [63]. The cleavage site may also lie outside the targeted complementary region, but this probability is very low [41]. *miR408* has multiple prediction targets, and *miR408*-mediated transcript cleavage is mainly located at the tenth and eleventh nucleotides (Appendix A). In some cases, *miR408*-mediated transcript cleavage site may also be located outside the target complementary region (Appendix A).

In *A. thaliana*, *miR408*-mediated transcript cleavage is relatively complex. The cleavage site is not only located at the 9th–10th nucleotides, 10th–11th nucleotides, 11th–12th nucleotides, but also outside the complementary region. For instance, *miR408* cleaved *LAC3* at the 9th-10th nucleotides, and cleaved *PLC*, *LAC12* and *UCC2* at the 10th–11th nucleotides at the 5′ end [16,20,36]. Abdel-Ghany and Pilon [36] showed that *miR408* cleaves *LAC12* at the 11th–12th and 14th–15th nucleotides at the 5′ end, respectively [37]. In addition, they demonstrated that *miR408* targeting *LAC13* has two cleavage sites, the most typical cleavage site is the 10th–11th nucleotide site, and another cleavage site is outside the complementary region [36]. But Song et al. [21] and Zhang et al. [37] reported that *miR408* cleaves *LAC13* at the 10th–11th nucleotides and 14th–15th nucleotides from the 5′ end, respectively. For *PLC,*
*miR408* cleaves at the 9th-10th nucleotides and 12th–13th nucleotides from the 5′ end [21].

In *O. sativa*, *miR408*-mediated transcript cleavage is different from that in *A. thaliana*. For instance, *miR408* cleaves *UCL8* and *LAC3-like1* at the 10th–11th nucleotides at the 5′ end [40]. The *PLC-like1*, *UCL30*, and *DSHCT* cleavage sites guided by *miR408* are all located at the 11th–12th nucleotides from the 5′ end [25]. In addition, *miR408* targets *PLC* with three cleavage sites, which are at the 10th–11th, 14th–15th, and 17th–18th nucleotides [38].

In *N. tabacum*, *miR408* cleaves *UCC-**like1* at the 10th–11th and 11th–12th nucleotides at the 5′ end [20]. In contrast to the interaction between *UCC-like1* and *miR408*, the *PLC-like1* and *LAC12-like1* cleavage sites guided by *miR408* are located at the 10th-11th nucleotides of the 5′ end [20]. In *T. aestivum,*
*miR408* cleaves *TaTOC-A1*, *TaTOC-B1*, and *TaTOC-D1* at the 10th-11th nucleotides from the 5’ end [20]. In *S. miltiorrhiza*, *miR408* cleaves *LAC3* at the 7th–8th and 10th–11th nucleotides from the 5′ end [41]. In contrast to the interaction between *LAC3* and *miR408*, the *LAC18* cleavage site guided by *miR408* is located at the 10th–11th nucleotides from the 5′ end [20].

In *Ipomoea batatas*, *miR408* cleaves *IbPCL* (*Plantacyanin*) directly at the 10th nucleotide from the 5′ end [44]. In contrast to the interaction between *IbPCL* and *miR408*, *IbKCS* (*3-ketoacyl-CoA synthase 4*) is a typical target gene of *miR408* that contains two mismatches within the *miR408* sequence. The *IbKCS* cleavage sites guided by *miR408* are located at the 5th and 10th nucleotides from the 5′ end, which are non-canonical and canonical cutting sites, respectively [44]. In *S. officinarum*, the *Diphenol oxidase laccase* cleavage sites guided by *miR408* are located at the second to third nucleotides from the 5′ end [64]. In *M. truncatula*, *m**iR408* cleaved *PLC* at the 10th–11th nucleotides from the 5′ end [41]. In *Dimocarpus longan*, the *m**iR408*-3p targeting *DlLAC12* has two cleavage sites, the most typical cleavage site is the 10th site AAG/AGG, and the other cleavage site is outside the complementary region [65].

All these results indicate that *miR408* cleavage sites for the target gene are different in different species, and the transcript cleavage is usually located at the 10th and 11th nucleotides from the 5′ end of *miR408*. In addition, the cleavage site of *miR408* on the target gene may be different in the same species.

## 4. The Roles of *miR408* and Its Targets in Plant Development

The expression of *miR408* is significantly affected by a variety of developmental and environmental conditions; however, its biological function is unknown [21]. It has been shown that *miR408* is important for regulating the growth and development of plant leaves, flowers, seeds and roots. Two models for plant biology research are *O. sativa* and *A. thaliana*. We summarized the roles of *miR408* and its targets in *O.sativa* and *A. thaliana* in Figure 4. In addition, we also summarized the main regulatory roles of *miR408* and its targets in plant growth and development (Appendix A).

### 4.1. Leaf Development

The shape of leaves is spectacularly diverse. *miR408* and its targets are involved in the regulation of the development of leaves, including leaf size, petiole length and leaf flag angle [20,21,22,23,24]. In *A. thaliana*, the function loss of *miR408* decreased leaf size [21], while the overexpression of *miR408* significantly increased leaf size [21,22,23]. The overexpression of *miR408* in *N. tabacum* also significantly increased leaf size [20]. In addition, the overexpression of *miR408* in *A. thaliana* significantly increased the petiole length [21]. The mechanism by which overexpression of *miR408* induces increased leaf area and petiole length is summarized as follows: *miR408* down-regulates target genes, which belong to *PLC*, *LAC*, *UCL* and *cupredoxin* (Figure 4q) [20], and then directly and/or indirectly modulates cytoplasmic growth and/or gibberellic acid (GA) biosynthesis, which promotes the increase of leaf size by enhancing cell expansion (Figure 4r,s) [21].

Physiological and morphological traits of the flag leaf angle play important roles in determining crop grain yield and biomass. The overexpression of *miR408* decreased the flag leaf angle in *O. sativa* and *T. aestivu* [20,24]. *TaTOC1s* (*TaTOC-A1*, *TaTOC-B1*, and *TaTOC-D1*), the targets of *miR408,* have been identified in *T. aestivu* [24]. The function loss of *TaTOC1s* significantly decreased the flag leaf angle in *T. aestivu*. Further research discovered that *miR408* regulates the angle of flag leaf by down-regulating its target genes, which belong to PLC, LAC and TOC1 family members (Figure 4i) [20,24].

In summary, *miR408* plays crucial roles in leaf development by regulating target genes encoding Cu binding proteins (including PLC, LAC, UCL and TaTOC1s family members) (Figure 4 and Appendix A). In addition, these target genes directly and/or indirectly regulate cytoplasmic growth and/or GA biosynthesis to promote cell expansion and increase leaf size and petiole length.

### 4.2. Flower Development

Flowers are important organs of plants, and the timing of flowering is not only an interesting topic in developmental biology, but it also plays a significant role in agriculture for its effects on the maturation time of seeds [24]. It has been confirmed that flower development, including flower initiation and flowering morphogenesis, is regulated by *miR408* and its targets (*TaTOC1s*, *UCL8* and *LAC*) [24,25,26].

The correct timing of the switch from vegetative to reproductive growth, namely the floral transition, is critical for reproductive success in flowering plants. In *T. aestivu*, the overexpression of *miR408* and the function loss of *TaTOC1s* (target genes of *miR408*) caused an earlier and shorter heading time than in the wild type (WT), but the overexpression of *TaTOC1s* led to later flowering [24]. These results suggested that the *miR408* functions in the wheat by mediating *TaTOC1s* expression. Further analysis revealed that *TaFT1*, the key gene involved in the flowering time, was up-regulated in both *miR408* overexpressing and *TaTOC1s* knockdown transgenic wheat, while the expression of *TaCO1* was reduced [25]. All these results suggested that the *miR408* regulation of wheat heading time and duration are dependent on its *TaTOC1s* mRNA cleavage activity and this enhances the expression of *TaFT1* [24,25,26].

*miR408* and its target gene *UCL8* also regulate flowering morphogenesis, including the flower size, pollen wall, stigmas, anthers, peduncle diameter and peduncle vascular bundles. The overexpression of *miR408* significantly increased the flower size in *A. thaliana* [21]. In *O. sativa,* the overexpression of *miR408* significantly increased the diameter of the peduncle, but with the function loss of *UCL8* it formed vigorous pollen with a higher germination rate [26]. In contrast, the overexpression of *UCL8* decreased the diameter of the peduncle in the main panicle, caused a thinner pollen intine, smaller stigmas, twisted anthers and an abnormal germination of pollen grains [25].

Further research discovered that *miR408* regulates flower size (Figure 4k), the pollen wall (Figure 4g), the peduncle diameter and peduncle vascular bundles (Figure 4h) by down-regulating its target genes *PLC*, *LAC* and *UCL8* [21,25,26]. Among these, *UCL8* regulates the production of vitamin B1 (VB1) components through the interaction with OsPKIWI, and leads to significant irregularities in pollen tube growth and pollination, which then affects the seed setting rate in *O. sativa* (Figure 4f) [26].

All these results indicate that *miR408* and its targets play active roles in flower development, the control of flowering morphogenesis, pollen germination and pollen tube growth, and ultimately affect the fertility and seed setting rate. Therefore, *miR408* and its targets are closely related to seed development and grain yield in plants.

### 4.3. Seed Development and Grain Yield

As an important complex agronomic trait, grain yield is dependent upon numerous factors such as seed size, seed weight and seed number.

Seed size is one of the most important factors affecting grain yield in plants. In *A. thaliana* and *N. tabacum*, the overexpression of *miR408* significantly increased seed size and seed weight compared with the WT [20,22]. In *O. sativa*, the overexpression of *miR408* and the function loss of *UCL8* also increased the seed size and 1000-grain weight, which was manifested by a substantial increase in seed length and width and a slight increase in thickness [20]. In contrast, the overexpression of *UCL8* in *O. sativa* significantly decreased the 1000-grain weight [25]. These observations indicate that the seed size regulated by *miR408* is highly conserved both in monocots and dicots.

Silique length is an important yield trait, which positively correlates with the number of seeds per silique and seed weight. In *A. thaliana*, the overexpression of *miR408* resulted in a significant increase in silique length and increased grain yield, but the overexpression of *PLC* and *LAC13* resulted in a significant decrease in silique length and a decreased grain yield [21]. In addition, the overexpression of *PLC* and *LAC13* also significantly increased the number of siliques in the inflorescence and increased grain yield [21]. Interestingly, the overexpression of *miR408* in *A. thaliana* also significantly reduced the seed coat color and root length (Figure 4p), and the mechanism remains unclear.

The number of grains per panicle is largely associated with the morphologies of the panicle, including the number of primary and secondary branches in the panicle. In *O. sativa*, *miR408* positively regulates grain yield by increasing the panicle branches and grain number through cleaving its target gene *UCL8* [25]. In *O. sativa*, the overexpression of *miR408* or the function loss of *UCL8* significantly increased the number of panicle branches and the number of grains per panicle, but the overexpression of *UCL8* dramatically reduced the number of panicle branches. In contrast, the overexpression of *UCL8* dramatically reduced the effective number of grains per panicle [25]. In conclusion, *miR408* negatively regulates *UCL8* to affect seed size, seed weight and seed number to ultimately control grain yield in *O. sativa*. Similarly, miR408 negatively regulates *PLC* and *LAC13* to affect seed size, seed weight, silique length and seed number to ultimately control grain yield in *A. thaliana*.

Further research discovered that *miR408* regulates seed size (Figure 4c,n), seed weight (Figure 4a,o), silique length (Figure 4s), panicle branching (Figure 4e) and grain number (Figure 4d) by regulating its targets and plastocyanin (PC). The target genes of *miR408* encode the blue Cu proteins associated with electron carrier activity in photosynthesis, which affects Cu homeostasis in the plant cell. Similarly, PC can transfer electrons from the cytochrome b6f complex to the photosystem I in photosynthesis [66]. *miR408* down-regulates these target genes and transports Cu to PC, thereby affecting photosynthesis and ultimately controlling yield-related traits, and promoting grain yield (Figure 4a,b,m).

### 4.4. Seed Germination

*miR408* not only controls seed morphogenesis, but also regulates seed germination [16]. The GA/ abscisic acid (ABA) ratio is critical for seed germination. In *A. thaliana*, the overexpression of *miR408* or the function loss of *PLC or PIF1* increased GA but decreased ABA content. In contrast, the function loss of *miR408* or overexpression of *PIF1* decreased GA and maintained ABA content. Further research demonstrated that *PIF1* binds to the *miR408* promoter and represses the accumulation of *miR408*, which then silences *PLC* after transcription, thereby forming a PIF1–miR408–PLC repression cascade. In addition, the PIF1–miR408–PLC cascade regulates germination by modulating the ratio of GA/ABA [16].

## 5. Functional Roles of *miR408* and Its Targets in Response to Stresses

Biotic and abiotic stresses greatly affect the growth and development of plants. In order to cope with these challenges, plants have evolved complex strategies to deal with these unfavorable environments and minimize damage as much as possible, so as to produce adaptive responses through physiological and morphological changes [63,67]. In recent years, an increasing number of studies have shown that in addition to regulating plant growth and development, *miR408* is also stress responsive in many plant species. Among the identified roles of *miR408* and its targets in response to biotic and abiotic stress, some have been verified by performing overexpression, T-DNA insertion, and RNAi experimentation. Here, we have summarized the regulating roles and modified expression of *miR408* in response to biotic and abiotic stress (Table 1 and Table 2).

### 5.1. Cold Stress

Low temperature is one of the most common environmental stresses which seriously affects the growth and development of plants [95]. Sunkar and Zhu [10] first discovered miRNAs in response to cold stress by Small RNA Sequencing. Then Ma et al. [22] verified the roles of *miR408* and its targets in response to cold stress through an overexpression technology.

In *A. thaliana* under cold stress, the expression of *miR408* increased under low temperature stress, while the expression of its target genes *cupredoxin* and *LAC3* decreased. In addition, the overexpression of *miR408* increased the survival rate, maximum quantum efficiency (Fv/Fm) values and chlorophyll fluorescence, while it decreased malondialdehyde (MDA), electrolyte leakage and luminescence [22]. In addition, an increased expression of *CSD1* and *CSD2* were also observed [22].

In *O. sativa*, the expression of pre-*OsmiR408* and *OsmiR408* increased under cold stress. As expected, the expression of its target genes (*plastocyanin genes*, Os01g53880 and Os09g29390) were down-regulated by cold treatment. The overexpression of *miR408* in *O. sativa* improved cold tolerance and it exhibited better growth than the WT [74]. In addition, the overexpression of *miR408* decreased ion leakage, and increased SOD activity and proline content [74]. Cuproproteins (such as *cupredoxin)* that are reduced with the overexpression of *miR408,* may increase the endogenous availability of Cu for other cuproproteins (such as Cu/Zn superoxide dismutases (CSDs)), and these cuproproteins are involved in the mediation response to abiotic stress [22]. It has also been proved that the CBF-independent nuclear protein, Tolerant to Chilling and Freezing 1 (TCF1), which is related to Blue-Cu-Binding Protein (BCB), can regulate lignin biosynthesis in *A. thaliana* [96]. The reduction of lignin deposition in the cell wall increases its permeability and enhances its elasticity, allowing it to adapt to the growth of ice crystals, which may reduce or prevent damage to dehydrated cells and cell walls [96]. In loss-of-function *TCF1* mutants and *BCB* knockouts there was a decreased lignin content and increased freezing tolerance.

All the results of these studies clearly show that *miR408* helps plants improve their tolerance to cold stress by regulating the genes related to Cu homeostasis, oxidative stress, and lignin biosynthesis.

### 5.2. Salinity Stress

Salinity stress is one of the main environmental stresses that limit plant growth and productivity [97]. Previous studies have discovered that *miR408* plays functional roles in *A. thaliana*, *S. miltiorrhiza*, *O. sativa* and *T. aestivum* under salinity stress [22,32,40,78].

In *A. thaliana* under salinity stress, the expression of *miR408* was up-regulated, while its target gene *cupredoxin* was down-regulated. In addition, the overexpression of *miR408* reduced oxidative stress by increasing the length of the main root and lateral root and increasing the expression of *CSD1* and *CSD2,* thereby enhancing the salinity tolerance of *A. thaliana* [22]. Similarly, when *S. miltiorrhiza* was exposed to salinity, the expression of *miR408* was up-regulated [78]. The overexpression of *miR408* promoted seed germination and reduced the accumulation of ROS under salinity stress by increasing the expression of antioxidative enzyme genes, i.e., *NbSOD*, *NbPOD*, and *NbCAT* and increasing their enzyme activities [78].

However, in *O. sativa* under salinity stress, the expression of *miR408* was down-regulated, while the expression of its target genes *helicase DSHCT* and *plastocyanin-like* were up-regulated [40]. *TaCLP1,* the gene target of *miR408*, plays a positive role in the response of *T. aestivum* to high-salinity. The overexpression of *TaCLP1* in yeast significantly increased cell growth under high salinity, but the mechanism remains unclear [32].

In summary, although increasing the expression of antioxidant enzyme genes is considered to be one of *miR408*’s strategies to improve plant salinity tolerance, many other mechanisms still need to be further studied. In particular, the up-regulation or down-regulation of *miR408* expression in different plants under salinity stress is inconsistent. The reason for these opposite results is not clear, indicating that different plants have different salt stress resistance mechanisms.

### 5.3. Drought Stress

Drought stress is a major constraint to agricultural productivity worldwide [98]. Using miRNA high-throughput sequencing, researchers have identified that the expression of *miR408* was significantly up-regulated or down-regulated under drought stress, as shown in Table 1. Further analysis showed that the up-regulation or down-regulation of *miR408* expression depends on the species, variety, tissue distribution and the degree of drought stress.

The response of *miR408* to drought stress varies in different species of plants. When exposed to drought stress, the expression of *miR408* was up-regulated in *Prunus dulcis* [68], *Prunus persica* [68], *C. arietinum* [75], *Hordeum vulgare* [77], *T. aestivum* [72], *Ipomoea campanulata* [69], *Jacquemontia Pentantha* [70] and *M. truncatula* [42]. On the contrary, when exposed to drought stress, the expression of *miR408* was down-regulated in *P. dulcis* [68], *Lycopersicon esculentum* [73], *Pisum sativum* [76], *A. thaliana* [22] and *I. Campanulata* [70].

The response of *miR408* to drought stress also depends on cultivars. When convolvulacee was exposed to drought stress, the expression of *miR408* was up-regulated in the tolerant wild *Ipomoea campanulata*, and down-regulated in the sensitive cultivated *J. Pentantha* [70]. Contrarily, another study illustrated that under drought stress, the expression of *miR408* was down-regulated in tolerant wild *I. campanulate*, and up-regulated in sensitive cultivated *J. Pentantha* [69]. In rice, during drought the expression of *miR408* was up-regulated in the fag leaves of the tolerant cultivars, Nagina 22 (N22) and Vandana, but down-regulated in the sensitive cultivars, Pusa Basmati 1 (PB1) and IR64 [39,71].

In addition, the responses of *miR408* to drought stress also depends on different tissues. In *T. aestivum* exposed to drought stress, *miR408* was down-regulated in root, but up-regulated in leaf [72].

The responses of *miR408* to drought stress also depends on the degree of drought. In *P. persica*, the expression of *miR408* was up-regulated under mild drought stress, while it was down-regulated under severe drought stress [68].

The functional roles of *miR408* and its target genes in response to drought stress are shown in Table 2. When *A. thaliana* was exposed to drought stress, the expression of *miR408* was up-regulated, while the expression of its target genes *cupredoxin*, *PLC* and *LAC3* were down-regulated. In addition, the overexpression of *miR408* increased the sensitivity to drought stress in *A. thaliana* [22]. On the contrary, a loss-of-function mutant of *miR408* had a decreased death rate and increased height under drought stress [22].

In *L. perenne*, the overexpression of *miR408* changed the leaf morphology and increased the antioxidant capacity, which in turn improved the ability of plants to cope with drought stress [93].

In *C. arietinum*, the overexpression of *miR408* showed a high ability to provide drought tolerance [75]. It is known that *Element Binding Protein 1A* (*DREB1A*) and *Dehydration-Responsive Element Binding Protein 2A* (*DREB2A*) are transcription factors involved in drought tolerance, and the overexpression of *miR408* increased the expression of DREB2A, DREB1A and *Rd17/29A*. Moreover, the expression of *PLC* (a target of *miR408*) was significantly decreased with the overexpression of *miR408*. Further research demonstrated that the overexpression of *miR408* leading to *PLC* transcript repression, caused the regulation of *DREB* and other drought responsive genes. All these results confirm that the *DREB1A* and *DREB2A* transcription factors and their target genes *RD17* and *RD29* provide plant tolerance allowing them to survive under drought stress [75].

In *O. sativa*, the overexpression of *miR408* decreased drought tolerance. In addition, after recovering from drought stress, the overexpression of *miR408* decreased the survival rate and increased water loss rates [74]. When exposed to drought stress, the tolerant cultivar exhibited a higher amount of transcription factor SPL9, higher ROS, lower Cu transporters (COPT1, COPT2, COPT4, COPT5, COPT7 and ATPases) and lower internal Cu levels than the sensitive cultivars PB1 and IR64 [71]. Further research showed that *miR408* was regulated by *OsSPL9* and directly interacted with the promoter via the GTAC motif [71]. In addition, *miR408* was up-regulated to decrease the transcript levels of *plastocyanin-like* in an ABA and Ca^2+^-dependent manner [39].

In total, the mechanisms of *miR408* in improving drought tolerance are summarized as follows (Figure 5): (1) Under drought stress, the internal Cu content of flag leaves decreases, which may be due to differences in Cu transporters. Then, lower Cu leads to the up-regulation of *miR408* via the transcription factor *OsSPL9*. Elevated *miR408* levels down-regulate target genes encoding Cu-containing proteins (PLC), and then cause cellular Cu to be preferentially distributed to PCs, thereby maintain PC activity and electron transport. The electron transport pathway is one of the main ROS production pathways. Finally, ROS promotes the closure of stomata in response to drought stress (Figure 5a) [39,71]. (2) *miR408* enhances the activities of SOD, CAT and POD, leading to a decrease in ROS accumulation, thereby improving drought tolerance (Figure 5b) [93]. (3) *miR408* down-regulates the target gene *PLC*, leading to an accumulation of Cu levels, and *DREB* levels increase when Cu is excessive. Subsequently, the *DREB2A* target gene *RD29B* is down-regulated, while the *DREB1A* and *DREB2A* transcription factor target genes *RD17* and *RD29A* are up-regulated. Finally, *DREB1A* and *DREB2A* t and their target genes *RD17* and *RD29* provide plant survival tolerance under drought stress [75].

### 5.4. Nutrient Deficiency

*miR408* plays important roles in regulating the response of plants to nutrient deficiencies. Nitrogen (N), phosphorus (P) and potassium (K) are the three most important macronutrients required for plant growth and crop yield. The expression of *miR408* was repressed in *A. thaliana* under N starvation, while the expression of its target genes *LAC* and *PLC* increased [81]. Similarly, the expression of *miR408* was repressed in *A. thaliana* under C, N, and S deficiency, while the expression of its target genes *LAC3* and *LAC13* increased [28]. When *Z. mays* was exposed to N deficiency, the expression of *miR408* was down-regulated, while the expression of its target gene *cupredoxin* and *SOD1A* were up-regulated [79,80]. *miR408* has different regulatory roles in different species of plants under P deficiency. In *A. thaliana* and *Glycine max*, *miR408* was significantly up-regulated under P deficiency [84,85]. However, in *T.aestivum*, *miR408* was down-regulated under P deficiency stress [86]. When *T.aestivum* was exposed to K deficiency, the expression of *miR408* was down-regulated [88]. In *Linum usitatissimum*, the expression of *miR408* was induced under excessive fertilization treatments [83]. In *Citrus sinensis*, B-deficiency induced the down-regulation of *miR408* and the up-regulation of *PLC*, *LAC3* and *LAC13* [82].

Although nutrients are essential elements required for the completion of plant growth and development [99], high concentrations of some mineral nutrients are harmful to plants. Cu plays essential roles in photosynthesis, oxidation responses and other physiological processes in plants [100]. Cu deficiency leads to an abridged photosynthetic electron transport, decreased pigment synthesis, and a disintegrated thylakoid membrane [101]. *miR408* interacts with genes encoding Cu-containing proteins such as cupredoxin, PLC, UCL and LAC, all of which belong to the phytocyanin family [52,60,102,103]. In addition, an increasing number of studies have shown that *miR408* is responsive to Cu deficiency in plants. It is inferred that *miR408* is the main regulator in Cu homeostasis.

In *A. thaliana*, the expression of *miR408* was up-regulated under Cu deficiency, while the expression of the target genes *cupredoxin*, *PLC*, *UCL*, *LAC3*, *LAC12* and *LAC13* were down-regulated [22,36]. The regulation of *miR408* under Cu deficiency was mediated by *SQUAMOSA promotor binding protein-like 7* (*SPL7*) by binding to the GTAC motifs of the *miR408* promoter [23]. Another identified transacting factor that directly mediates *miR408* expression via promoter area binding is HY5 (elongated hypocotyl 5). In *A. thaliana*, by directly interacting with the promoter through GTAC and G-box motifs, SPL7 and HY5 act in coordination to transcriptionally regulate *miR408*, which results in the differential expression of *miR408* and its target genes in response to changing light and Cu conditions [37]. This regulation makes sense, since in the light required for photosynthetic growth, which has a high demand for Cu, HY5 mediates gene expression.

In addition to the regulation of lignin biosynthesis, *miR408* may regulate the iron deficiency response in complex ways [18]. In *A. thaliana*, when exposed to Fe deficiency, the expression of *miR408* was up-regulated in the WT, while its target genes *LAC3*, *LAC12*, *LAC13* and *plantacyanin* (*ARPN*) were down-regulated. The overexpression of *miR408* leads to enhanced Fe sensitivity [18]. It increased the expression of lignification-related genes *F6′H1′*, *B-GLU23* and *LAC17* under Fe deficiency, and at the same time decreased the expression *bHLH29* and its target genes *LAC3*, *LAC12*, *LAC13* and *PLC*.

Zinc (Zn) is an essential micronutrient in plants and *miR408* is an important regulator of the response to Zn deficiency in plants. In *Sorghum bicolor*, *miR408* in leaves was significantly up-regulated by Zn deficiency, while the expression of its target gene *plantacyanin* was decreased [87]. However, the regulatory mechanism of *miR408* in Zn deficiency still needs further study.

### 5.5. Other Abiotic Stresses

The expression of *miR408* not only changes under cold, salinity, drought and nutrient stresses, but also alters under methyl viologen (MV)-induced oxidative, osmotic and heavy metals stresses.

*miR408* plays a positive function in enhancing the oxidation tolerance of *A. thaliana*. The expression of *miR408* in *A. thaliana* was up-regulated under oxidative treatment, while its target gene *PLC* was repressed [22]. In addition, when exposed to MV-induced oxidative stress, the overexpression of *miR408* increased the biomass and root length, while the function loss of *miR408* decreased the root length in *A. thaliana*. Further research showed that elevated levels of *miR408* can increase tolerance to MV-induced oxidative stress by reducing the accumulation of ROS [22]. In contrast, the overexpression of *miR408* reduces the tolerance of plants to osmotic stresses. Under osmotic stress, in *A. thaliana* the overexpression of *miR408* showed growth retardation and a decreased fresh weight, while the function loss of *miR408* showed better growth and an increased fresh weight [22].

Interestingly, when exposed to metal stresses, the expression of *miR408* changes with time and varies in different organs. In *T. aestivum* under cadmium (Cd) stress, the expression of *miR408* in the leaves was down-regulated, reaching their lowest expression level at 12 h, and then reaching a relatively high expression level at 24 h. The expression of its target gene, *chemocyanin-like protein,* was down-regulated in leaves, gradually decreasing at 6 h, increasing at 12 h, and then slightly decreasing at 24 h. In addition, *miR408* was up-regulated and maintained extremely high expression levels at all time points in the roots (6, 12, 24 and 48 h). The expression of *chemocyanin-like protein* in roots was up-regulated after 24 and 48 h, reaching the highest expression at 48 h [89]. However, in *O. sativa*, *miR408* family members are up-regulated under Cd stress and play important roles in reducing Cd accumulation by down-regulating the expression of candidate genes, such as *bZIP*, *ERF*, *MYB*, *SnRK1* and *HSPs* [29].

In addition, using miRNA arrays for miRNA analysis, researchers found that *miR408* family members were up-regulated in *O. sativa* under Arsenic (As) (III) and As (V) stress [30]. *miR408* was strongly inhibited in *Phaseolus vulgaris* leaves or roots under manganese (Mn) toxicity [90]. In addition to responding to Cu deficiency, *miR408* also responds to high Cu stress [43].

Overall, the results demonstrate a significant involvement of *miR408* in abiotic stress responses, emphasizing the central function of *miR408* in plant survival [22], but the detailed mechanisms are still poorly understood and further studies are needed.

### 5.6. Biotic Stress

*miR408* not only plays important roles in the tolerance to various abiotic stresses, but also plays vital roles in the tolerance to biotic stresses such as LPS stress, *Puccinia*
*striiformis f. sp. Tritici* (Pst) infection, *Puccinia graminisfsptritici* infection and *Rhizoctonia solani* infection. When *A. thaliana* was exposed to LPS stress, the expression of *miR408* was up-regulated, while its target gene *PLC* was inhibited [31]. Wheat stripe rust caused by Pst is one of the most destructive diseases in the world, and it can greatly reduce or even completely destroy *T. aestivum* yields in epidemic years [32]. *TaCLP1* is the target of *miR408*, which positively regulate stripe rust resistance in *T. aestivum* [32]. Compared with the WT control, the function loss of *TaCLP1* in *T. aestivum* decreased its resistance to stripe rust, showing some fungal sporulation, larger necrotic areas and a larger hyphal length [32]. *Puccinia graminisfsptritici* is a rust fungus causing serious disease to *T. aestivum* [91]. Gupta et al. [94] reported that *miR408* in *T. aestivum* was involved in the defense response to stem rust infection. During Pst infection, the up-regulation of *miR408* may trigger the lignin biosynthetic pathway as a hypersensitive response (HR) response to prevent epidermis rupture. Sheath blight disease, caused by *Rhizoctonia solani*, is one of the most destructive *O. sativa* diseases, resulting in an estimated yield loss of up to 50% [92]. In *O. Sativa*, a late responsive *miR408* might be involved in *R. solani* infection in rice plants. In addition, the expression of *miR408* was up-regulated under *R. solani* infection [104].

As shown in Table 1 and Table 2, *miR408* is involved in various biotic and abiotic stresses, but the detailed regulation mechanism of *miR408s* in response to stresses has not yet been fully explored. Existing studies have shown that *miR408* has a great relationship with the antioxidant system, but no studies have shown that *miR408* directly regulates the expression of specific genes involved in the antioxidative system. Ma et al. [22] proposed that increasing the expression of *miR408* can reduce the level of ROS and regulate the target genes encoding Cu-containing proteins, thereby increasing the endogenous availability of Cu for other Cu proteins involved in response to abiotic and biotic stress. In addition, Zhang et al. [37] showed that *miR408* is an important component part of the HY5–SPL7 gene network and can mediate a coordinated response to light and Cu. Peñarrubia et al. [105] proposed that the interaction between Cu homeostasis and plant hormones (mainly ABA and ethylene) is related to abiotic stress. Therefore, the mechanism of *miR408* in response to stress may go beyond a simple interaction between a single *miR408* and its target genes. More attention should be paid to the interaction of *miR408* and target genes with other transcription factors and hormones.

## 6. Conclusions

Evolutionary analysis shows that the *miR408* family is an ancient and widely distributed miRNA family, and bryophytes might be the evolutionary ancestors of the *miR408* family. There are fewer *miR408* species in ancient and lower plants, and more *miR408* species in higher dicotyledonous plants, indicating that *miR408* evolved gradually from bryophytes and may play an important role in the evolution of plants to dicotyledonous plants. It may be that the emergence and changes of *miR408* led to the evolution of various higher plants. A number of studies have shown that *miR408* plays significant roles in regulating plant growth, development and stress response. *miR408* affects the growth and development of plant leaves, flowers, seeds and roots by down-regulating the target genes encoding blue Cu proteins; by transporting Cu to PC, it regulates photosynthesis and ultimately promotes the increase of grain yield. In addition, *miR408* can also enhance the content of antioxidants in cells, improve the antioxidant capacity of plants, and thereby improve the tolerance to various stresses. All these results greatly expand our understanding of the evolution and functional roles of *miR408* and its targets in plants.

## Figures and Tables

**Figure 1 ijms-23-00530-f001:**
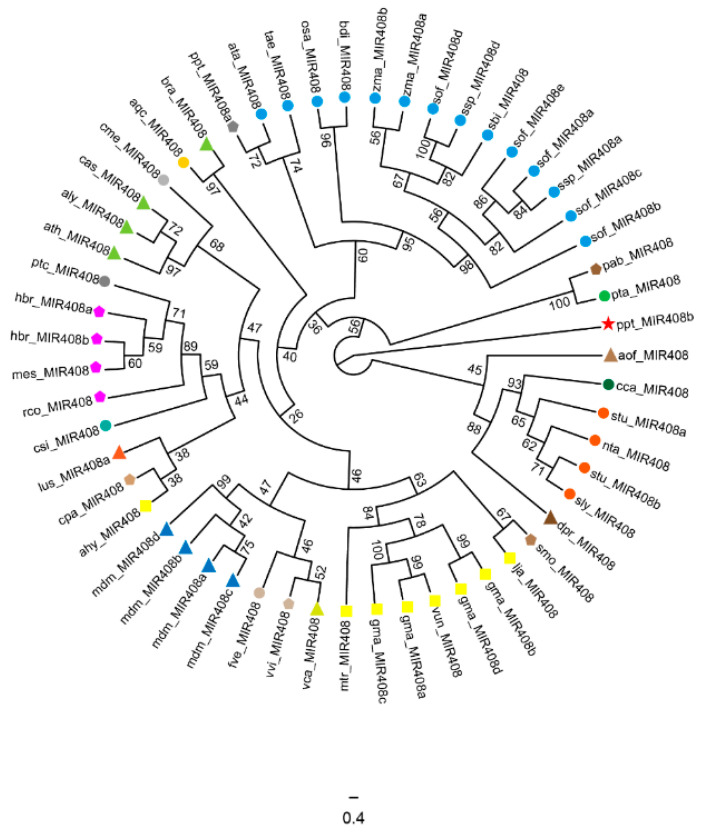
Phylogenetic relationships between all the 55 *miR408* precursors (*MIR408*) from different plants. The colored branch shows the same family species. The dendrogram has been constructed using IQtree software by the Maximum-Likelihood method with 5000 bootstraps. According to IQ-Tree model evaluation, the best fitting model of the *miR408* precursor sequence is TVMe+I+G4, and the best model of the *miR408* mature sequence is JC. The Bryophyta *Physcomitrella patens* is the root of the evolutionary tree. aly, *Arabidopsis lyrata*; aqc, *Aquilegia caerulea*; ata, *Aegilops tauschii*; ath, *Arabidopsis thaliana*; ahy, *Arachis hypogaea*; bdi, *Brachypodium distachyon*; bra, *Brassica rapa*; cca, *Cynara cardunculus*; cme, *Cucumis melo*; cpa, *Carica papaya*; csi, *Citrus sinensis*; dpr, *Digitalis purpurea*; gma, *Glycine max*; hbr, *Hevea brasiliensis*; lja, *Lotus japonicus*; lus, *Linum usitatissimum*; mdm, *Malus domestica*; mes, *Manihot esculenta*; mtr, *Medicago truncatula*; nta, *Nicotiana tabacum*; osa, *Oryza sativa*; pta, *Pinus taeda*; ptc, *Populus trichocarpa*; ppt, *Physcomitrella patens*; rco, *Ricinus communis*; sbi, *Sorghum bicolor*; sof, *Saccharum officinarum*; stu, *Solanum tuberosum*; smo, *Selaginella moellendorffii*; tae, *Triticum aestivum*; vvi, *Vitis vinifera*; vun, *Vigna unguiculata*; zma, *Zea mays*; ssp, *Saccharum ssp*; vca, *Vriesea carinata*; sly, *Solanum lycopersicum*; fve, *Fragaria vesca*; pab, *Picea abies*; cas, *Camelina sativa*; aof, *Asparagus officinalis*.

**Figure 2 ijms-23-00530-f002:**
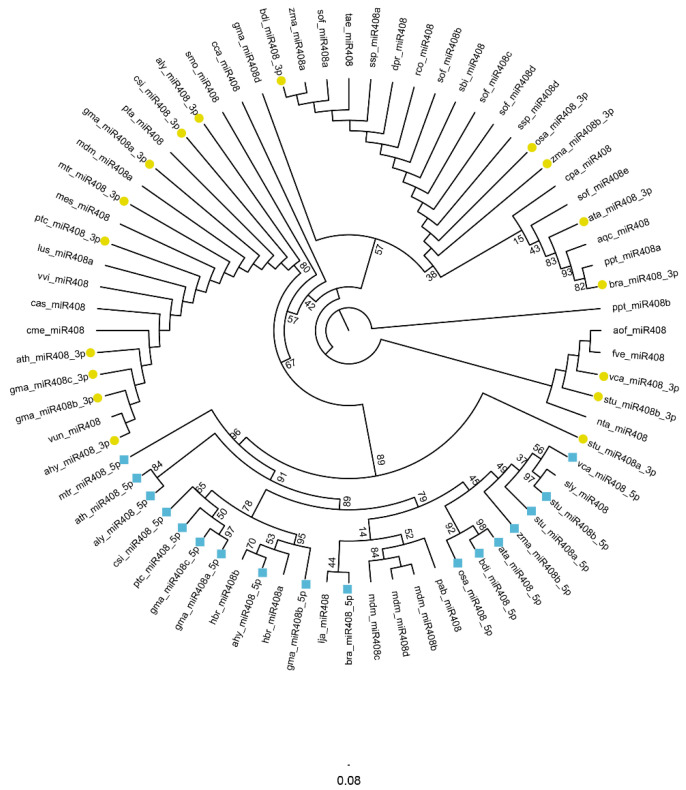
Phylogenetic relationships between all 72 mature *miR408* sequences (*miR408*) from different plants. The yellow circle and blue square represent the *miR408* members formed from the 3p arm and the 5p arm, respectively. The dendrogram is constructed using IQtree software through the Maximum-Likelihood method using 5000 bootstraps. According to IQ-Tree model evaluation, the best fitting model of the *miR408* mature sequence is JC. The bryophyta *Physcomitrella patens* is the root of the evolutionary tree.

**Figure 3 ijms-23-00530-f003:**
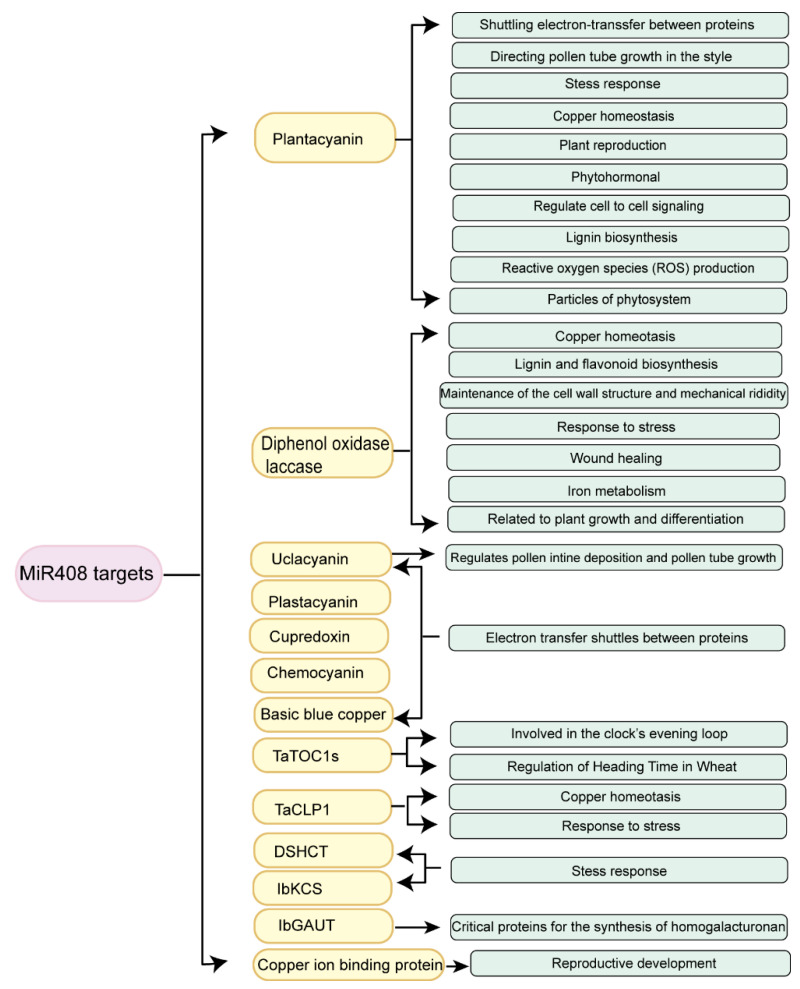
The function of *miR408*-regulated target genes in plants. Yellow, target genes; green, functions of the target genes.

**Figure 4 ijms-23-00530-f004:**
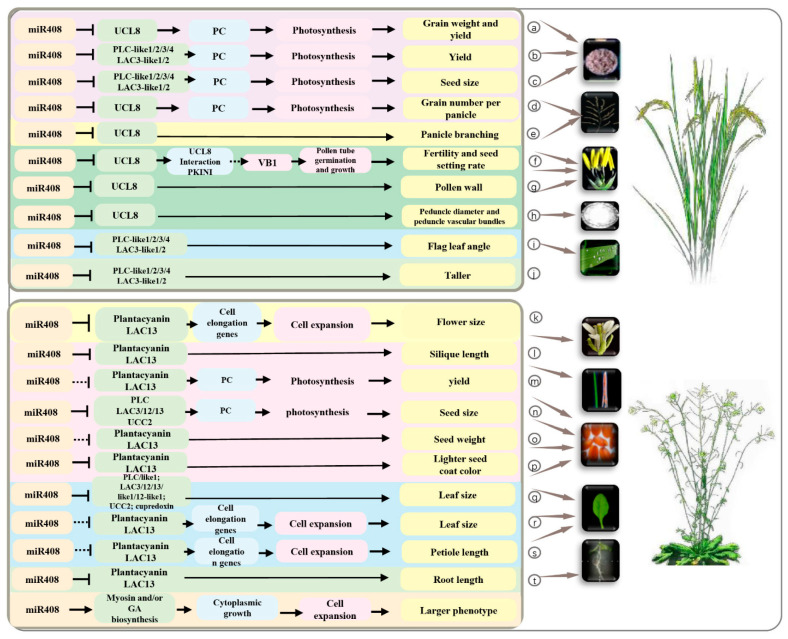
The roles of *miR408* and its targets in plant development. *miR408* (highlighted in light pink); target genes (highlighted in green); agronomic traits (highlighted in yellow). The arrow and nail shapes indicate positive or negative regulation, respectively. *LAC*, *laccase*; *PLC*, *plantacyanin*; *UCL8*, *Uclacyanin-like protein 8*; PC, plastocyanin; VB1, Vitamin B1.

**Figure 5 ijms-23-00530-f005:**
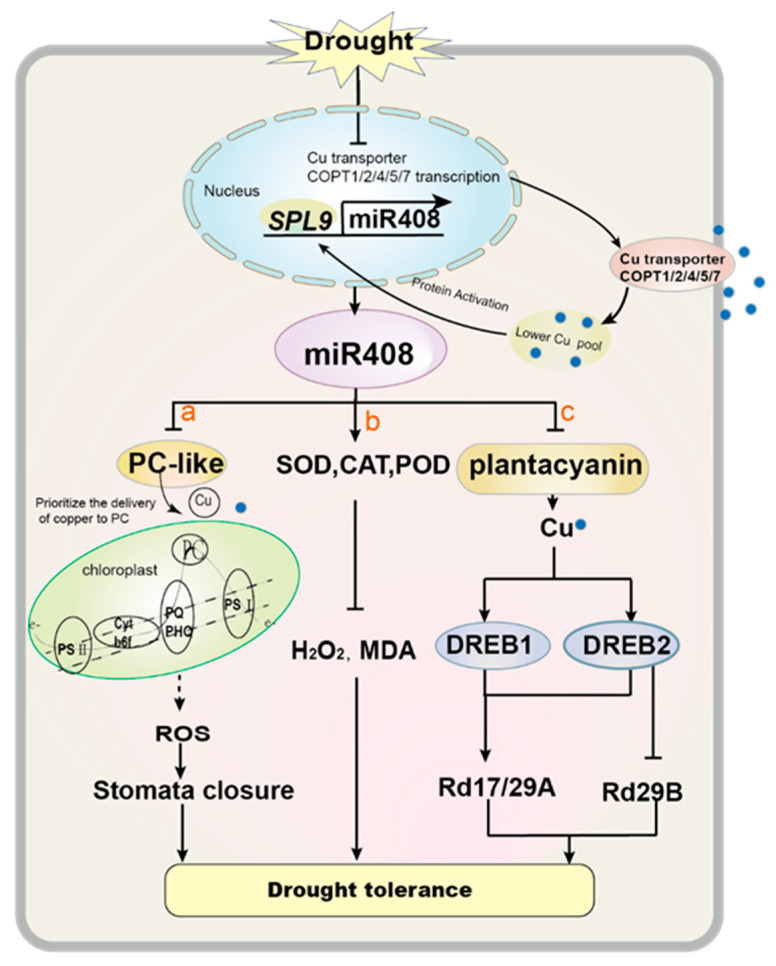
The potential regulatory mechanism model of *miR408* and its targets in plant tolerance to drought stress. In brief, drought stress decreases internal copper (Cu) levels by inhibiting the expression of Cu transporters. Then a lower concentration of Cu activates the expression of *SQUAMOSA promoter binding protein (SBP)-*like 9 (*SPL9*) and then up-regulates *miR408*. a, *miR408* targets several Cu containing proteins. These targets include several member proteins containing *plantacyanins*/*plastocyanin-like domains*. *miR408* down-regulates target genes, saves Cu for plastocyanin, and maintains a stable plastocyanin level, thereby increasing reactive oxygen species (ROS). At the same time, high concentrations of ROS promote stomata closure, thereby enhancing drought tolerance [75]. b, *miR408* leads to higher activities of superoxide dismutase (SOD), catalase (CAT) and peroxidase (POD), and leads to lower hydrogen peroxide (H_2_O_2_) and malondialdehyde (MDA). c, *miR408* down-regulates the target gene *PLC* and then increases Cu accumulation. Then the *DREB* level is increased when there is excess Cu. Subsequently, the target gene *RD29B* of *DREB2A* is down-regulated. The target genes of *DREB1A* and *DREB2A, RD1 7*and *RD29A* are up-regulated. All these results confirm that *DREB1A* and *DREB2A* transcription factors and their target genes *RD17*and *RD29* provide plants with tolerance for survival under drought stress. Dark blue: transcription factors; yellow: *miR408* target genes.

**Table 1 ijms-23-00530-t001:** Roles of *miR408* and its targets in abiotic and biotic stress in plants.

Stresses	Species	miRNA	Target Genes	References
*ABIOTIC*				
Mild drought	* Prunus persica *	*miR408*	Plastocyanin,	[68]
			Copper ion	
			binding protein	
	*Prunus dulcis*	*miR408*	No	
Severe drought	* Prunus persica *	*miR408*	Plantacyanin	
	*Prunus dulcis*	*miR408*	No	
	*Interspecific Prunus persica–Armeniaca Prunus dulcis*	*miR408*	No	
Drought	*Convolvulacee*	*miR408*	No	[69]
	(*tolerant wild Ipomoea*			
	*campanulata*)			
	*Convolvulacee*	*miR408*	No	
	(*sensitive Cultivated*			
	*Jacquemontia Pentantha*)			
	*Convolvulacee*	*miR408*	No	[70]
	*(tolerant wild Ipomoea*			
	*campanulata* *)*			
	*Convolvulacee*	*miR408*	No	
	*(sensitive Cultivated*			
	*Jacquemontia Pentantha* *)*			
	* Oryza sativa *	*miR408*-3p	Plantacyanin,	[39,71]
	(*tolerant Cultivars N22*, *dana*)		plastocyanin-like	
			domain,	
			containing	
			proteins	
	* Oryza sativa *	*miR408*-3p	No	
	(*sensitive cultivars PB1*, *IR64*)			
	*Triticum aestivum*	*miR408*	No	[72]
		(root)		
		*miR408*	No	
		(leaf)		
	* Lycopersicon esculentum *	*miR408*a-3p	No	[73]
	(*sensitive*)			
	* L.esculentum var. *	*miR408*	No	
	* Cerasiforme * (*tolerant*)			
	*Pisum sativum*	*miR408*	No	[72]
	*Oryza sativa*	*miR408*	*plastocyanin genes* (Os01g53880 and Os09g29390)	[74]
	*Arabidopsis thaliana*	*miR408*	Cupredoxin	[22]
			Plantacyanin	
			LAC3	
	*Cicer arietinum*	*miR408*	No	[75]
Water	*Ipomoea campanulate*	*miR408*	Plantacyanin	[70]
deficit	*(tolerate)*			
	*Jacquemontia Pentantha*	*miR408*		
	(*sensitive*)			
	*Pisum sativum*	*miR408*	P1B-ATPase	[76]
	*Medicago truncatula*	*miR408*	Plantacyanin	[42]
Dehydration	*Hordeum vulgare*	*miR408*	Plantacyanin	[77]
Salinity	*Oryza sativa*	*miR408*	DSHCT	[40]
			Plastocyanin-like	
	*Salvia miltiorrhiza*	*miR408*	No	[78]
	*Arabidopsis thaliana*	*miR408*	Cupredoxin	[22]
			Plantacyanin	
			Uclacyanin	
			LAC3	
Cold		*miR408*	Cupredoxin	
			Plantacyanin	
			Uclacyanin	
			LAC3	
Osmotic		*miR408*	Cupredoxin	
			Plantacyanin	
			Uclacyanin	
			LAC3	
Oxidative		*miR408*	Cupredoxin	
			Plantacyanin	
			Uclacyanin	
			LAC3	
Low dose	*Oryza sativa*	*miR408*	DSHCT	[27]
rate γ-ray				
High dose		*miR408*		
rate γ-ray				
*Nutrient deprivation*				
	*Zea mays*	*miR408*	Cupredoxin	[79,80]
Nitrogen			SOD1A	
deficiency	*Arabidopsis thaliana*	*miR408*	Laccase	[81]
			Plantacyanin	
Carbon, nitrogen,		*miR408*	LAC3	[28]
and sulfur			LAC13	
deficiency				
Copper deficiency		*miR408*	LAC3	[22,36]
			LAC12	
			LAC13	
			Plantacyanin	
Iron deficiency		*miR408*	LAC3	[18]
			LAC12	
			LAC13	
			Plantacyanin	
Boron deficiency		*miR408*	Plantacyanin	[82]
			LAC3	
			LAC13	
			Cu/Zn SODs	
			(CSDs)	
Excess fertilizer	*Linum usitatissimum*	*miR408*	No	[83]
Phosphorus deficiency	*Arabidopsis thaliana*	*miR408*	No	[84]
	*Glycine max*	*miR408*	No	[85]
	*Triticum aestivum*	*miR408*	No	[86]
Zinc deficiency	*Sorghum bicolor*	*miR408*	Plantacyanin	[87]
Potassium	*Triticum aestivum*	*miR408*	No	[88]
deficiency				
*Heavy metals*				
*Cadmium*	*Triticum aestivum*	*miR408* (12h, leaves)	Chemocyanin-like protein	[89]
		*miR408* (24h, leaves)		
		miR408 (6, 12, 24 and 48h, roots)		
	*Oryza sativa*	*miR408*	bZIP, ERF,	[29]
			MYB, SnRK1	
			and HSPs	
Arsenate and	*Oryza sativa*	*miR408*	No	[30]
arsenite				
Manganese	*Phaseolus vulgaris*	*miR408*	No	[90]
*BIOTIC*	*Triticum aestivum*	*miR408*	Plantacyanin	[91]
Puccinia graminis				
f.sp. tritici				
Rhizoctonia solani	*Oryza sativa*	*miR408*	No	[92]
Lipopolysaccharide	*Arabidopsis thaliana*	*miR408*	Plantacyanin	[31]

Green: increased *miR408* abundance; red: decreased *miR408* abundance. DSHCT, DOB1/SK12/helY-like DEAD-box Helicase; *LAC*, *laccase*.

**Table 2 ijms-23-00530-t002:** Modified expression of plant *miR408* and its targets in response to abiotic and biotic stress.

Stresses	Species	Approach	Phenotype	References
Cold	*Arabidopsis thaliana*	*miR408* overexpression	More tolerant to cold tolerant,	[22]
			Lower electrolyte leakage,	
			Higher Fv/Fm value,	
			Lower MDA,	
			Higher chlorophyll	
		T-DNA *miR408* mutant	Enhanced cold sensitivity,	
			Enhanced electrolyte leakage,	
			Lower Fv/Fmvalue,	
			MDA were elevated,	
	*Oryza sativa*	*miR408* overexpression	Lower chlorophyll	[74]
			longer shoots and roots,	
			Lower ion leakage,	
			Enhanced SOD activity,	
			Enhanced proline content	
Salinity	*Arabidopsis thaliana*	*miR408* overexpression	Root development was better,	[22]
			Lower ROS	
		T-DNA *miR408* mutant	Inhibited root development,	
			Enhanced ROS	
	* Triticum aestivum *	TaCLP1 overexpression	In yeast enhances cell tolerance	[32]
	*Salvia miltiorrhiza*	*miR408* overexpression	Improved root growth,	[78]
			Significantly higher fresh weights,	
			Higher germination rates,	
			Lower growth inhibition,	
			Reduced ROS Accumulation,	
			Lower accumulations of H_2_O_2_,	
			Higher POD, SOD, CAT activities,	
			Lower levels of O_2_^−^and H_2_O_2_	
Oxidative	*Arabidopsis thaliana*	*miR408* overexpression	More tolerant to oxidative stress,	[22]
			Higher biomass,	
			Higher total root length,	
			Increased CSD1, CSD2,	
			CCS1, GST-2U25 and SAP12	
		T-DNA *miR408* mutant	Lower total root length,	
			CSD1, CSD2, CCS1,	
			GST-U25 and SAP12 reduced	
Drought	*Arabidopsis thaliana*	*miR408* overexpression	Retarded growth,	[22]
			Lower FW,	
			Death rate higher	
		T-DNA miR408 mutant	Grew better,	
			Higher height,	
			Death rate lower	
	*Cicer arietinum*	*miR408* overexpression	High stress tolerance,	[75]
			Lower height,	
			Increasing number of leaves,	
			BHLH23 down-regulated,	
			ERF/AP reduced expression,	
			DREB2A/1A genes were increased,	
			Rd17 and Rd29a were increased,	
			Rd22 up-regulated	
	*Lolium perenne*	*miR408* overexpression	Narrower leaves of similar length,	[93]
			Less vein number,	
			More closely folded leaves,	
			Greener,	
			Higher chlorophyll,	
			More bristle-like trichomes on the	
			leaf surface,	
			Relatively smaller and more sunken	
			stomata,	
			Lower stomatal conductance,	
			Less tissue damage,	
			Higher leaf RWC,	
			Lower water loss rate,	
			Higher leaf electrolyte leakage (EL),	
			Higher activities of SOD, CAT and	
			POD,	
			Lower accumulation of H_2_O_2_ and	
			MDA	
Osmotic	*Arabidopsis thaliana*	*miR408* overexpression	Retarded growth,	[22]
			Lower FW	
		T-DNA *miR408* mutant	Grow better,	
			Higher height	
Copper deficiency	* Triticum aestivum *	TaCLP1 overexpression	Higher tolerance to copper deficiency	[32]
Iron deficiency	*Arabidopsis thaliana*	T-DNA *miR408* mutant	Lower chlorophyll-a content,	[18]
			Lower lignin content,	
			*LAC3*, *LAC12*, *LAC13* and	
			plantacyanin (*ARPN*) up-regulated,	
			Lower lignification-related genes (F6’H1’, CCR1, B-GLU23, LAC17),	
			Lower *bHLH39*,	
			Higher phenoloxidase activity,	
			Higher H_2_O_2_ levels,	
			Lower *MCO3* and *CAT2*	
		*miR408* overexpression	Lower chlorophyll-a content,	
			Lower lignin content,	
			*LAC3*, *LAC12*, *LAC13* and	
			*plantacyanin* (*ARPN*) down-regulated,	
			lignification-related genes (F6’H1’,	
			B-GLU23, LAC17) significantly	
			increased,	
			Lower *FIT*,	
			Lower H_2_O_2_ levels,	
			lower *MCO3* and *CAT2*	
		Wild type	Higher Copper levels,	
			*miR408* expression decreased,	
			*LAC3*, *LAC12*, *LAC13* and	
			*plantacyanin* (*ARPN*) up-regulated,	
			Lower FRO2, FRO3, IRT1, COPT2,	
			Lower phenoloxidase activity,	
			Lower ferroxidase activity,	
			Higher H_2_O_2_ levels,	
			Higher lignin staining of vascular	
			cylinder,	
			Lignification of the vascular bundles	
			was more evident in the aerial part	
Puccinia striiformis f. sp. tritici	*Triticum*	RNAi TaCLP1 mutant	Decreased stripe rust resistance	[94]
	*aestivum*			

The same color symbol represents the same abiotic or biotic stress. MDA, malondialdehyde; ROS, Redox and reactive oxygen species; POD, peroxidase; SOD, superoxide dismutase; CAT, catalase; SODs, Cu/Zn superoxide dismutase CSD1 and CSD2; CCS1, copper chaperone; GST-U25, glutathione-S-transferase; SAP12, stress-associated regulatory protein fresh weight; BHLH23, transcription factor; ERF/AP2, Aptela2/Ethylene Response Factors; DREB, Dehydration-Responsive Element Binding Protein; bHLH39 and FIT transcriptional activators; MCO3, ascorbate oxidase; CAT2, catalase; FRO2, Ferric Reductase 2; IRT1, Iron Regulated Transporter 1.

## Data Availability

Not applicable.

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
