# Peer review of "The Evolution and Functional Roles of miR408 and Its Targets in Plants"

_ijms, 2022, doi:10.3390/ijms23010530_

Round 1

Reviewer 1 Report

In the submitted review, Gao and associates tried to summarize current research about miR408 in plants and represent its evolution and roles in plant development, growth and stress responses. The review mainly summarizes miR408 overexpression studies and the effect of OE on plants under normal and stress conditions. However, in my opinion, they failed in their mission to review and describe the roles of miR408.

The article mainly discusses the effect of miR408 overexpression (OE) on plant growth, development or stress resistance. However, OE under constitutive promoter does not show the role of miR but of its targets. During OE (especially under constitutive promoter), miR expresses in not the right places, wrong timing, and huge amounts that usually do not exist in plants. Consequently, miR OE cannot reveal the actual role of miR in various processes, and miR OE results in plenty of artifacts. miR OE allows to reveal the roles of its targets. To discuss the role of miR408, the authors should discuss miR408 knockout or silencing studies. Only one study, in which OE and knockout plants were used, was cited and mainly discussed OE effects in this research. At least three additional studies exist in which CAS9 mutant and amiR plants were used. These articles were not cited even.

Additionally, in my opinion, the authors should improve their discussion about the evolution of miR408 in the plant kingdom. The review mainly discusses the agricultural aspects of miR408 OE and more suits for submission to plant/agriculture journal (even in such case, the article's name should be changed because the article does not review miR408 roles but of its targets). Consequently, this review should be significantly improved.

1. miR OE does not show miR role but of its targets.
2. Additional studies that reported the roles of miR408:

Front Plant Sci.
2017; 8: 1598. doi: 10.3389/fpls.2017.01598
Plant Cell. 2021 Jul 2;33(5):1506-1529. doi: 10.1093/plcell/koab060 (STTM silencing)

3. Lines 40-42: In my opinion, it is good to give the stage for plant researchers also. It is
good to mention the works of D. Baulcombe, for example.

4. Lines 43-45: You are writing about conserved miRs. It is good to mention moss miRs
as the sample for conserved miRs: works of Axtell or W. Frank. Better to cite original
research and not reviews in this case.

5. Line 47: miR OE shows the role of its targets

6. Line 70: ref 9,17-it is not enough to cite only these two papers if you talk about “30
species”.

7. Lines 75-78: I think it is a place to say something about the difference in the number of
precursor and mature miRNA sequences. Is some precursor encodes for several mature
sequences?

8. Lines 80-90: Does miR408 exist in the non-plant kingdom? Does every species have
the same number of miR408 (please discuss the various size of miR family in various
species)?

9. Does miR408 exist in algae? If yes, please add it to your analysis and discuss it.

10. Lines 95-102: Does it important to talk about precursors? Is it important that they are
different (At the end, it is mature miRNA that is very similar to others)?

11. Lines 112-117: The number of miRs in comparison to P. patens, the evolution of the
miR408 family in monocots and dicots?

12. Fig. 1: check words in the fig. for ex: “trnsfer” to transfer, etc. Is it really a blue colour?

13. Lines 132-133: please mention that it is a prediction. I do not know any miR that
someone showed and validated such an amount of targets.

14. Line 200: ref 5 shows it in two places opposite to ref 21.

15. Then you are talking about OE- miR OE does not show miR role but of its targets.
Recently, a nice paper was published in Plos genetics that explained it very clear, but
of course, other people wrote it years ago (for ex. the works of T. Arazi).

16. Lines 284-285: references about stomata and ROS.

17. Table 1: please add column OE/DR overexpression and down-regulation, silencing,
knockout

18. Stresses: under stress conditions, many genes/proteins/miRs are differently affected in
plants. It can be a secondary/third effect, not necessary that change in expression of
some miR/gene contributes to resistance or sensitivity (it should be validated in mutant
plants).

19. Line 459: salt stress is complex stress, and includes several primary and secondary
stresses, and affects plant at all levels, so to mention only water uptake is not proper.
Especially that salt stress affects the roles of miR408 targets.

20. Line 478: better antioxidant system? Is it a car? Other, more effective, etc.

21. Line 488: these enzymes are not the targets of miR408; it can be secondary/third effects.

22. Line 492-494: too speculative.

23. Line 577: expression to expressed. Between?

Best regards

Author Response

Dear editors and reviewers, 
Thank you very much for your letter with regard to our manuscript (ID: IJMS-146078). We definitely appreciate your expert comments and constructive suggestions, we have revised the manuscript point by point according to your comments, all specific items have been answered or addressed and the revisions in the text have been highlighted with red color. In addition, we have used editing services from Elsevier to decrease language mistake and improve the quality of the manuscript. In the revised manuscript, we have provided more comprehensive analysis of the present progresses and developed a more detailed discussion. I hope that our revision will enable the manuscript to meet the accepted standards of IJMS this time.

If you have any question about the revised manuscript, please don’t hesitate to contact us as soon as possible. Once again, we acknowledge your expert comments and constructive suggestions, which are valuable for improving the quality of our manuscript. 
Best Regards. 
Sincerely yours, 
Jie Xiong 

Replies to reviewers

Reviewer 1

Major point:

Q1: In the submitted review, Gao and associates tried to summarize current research about miR408 in plants and represent its evolution and roles in plant development, growth and stress responses. The review mainly summarizes miR408 overexpression studies and the effect of OE on plants under normal and stress conditions. However, in my opinion, they failed in their mission to review and describe the roles of miR408.The article mainly discusses the effect of miR408 overexpression (OE) on plant growth, development or stress resistance. However, OE under constitutive promoter does not show the role of miR but of its targets. During OE (especially under constitutive promoter), miR expresses in not the right places, wrong timing, and huge amounts that usually do not exist in plants. Consequently, miR OE cannot reveal the actual role of miR in various processes, and miR OE results in plenty of artifacts. miR OE allows to reveal the roles of its targets. To discuss the role of miR408, the authors should discuss miR408 knockout or silencing studies. Only one study, in which OE and knockout plants were used, was cited and mainly discussed OE effects in this research. At least three additional studies exist in which CAS9 mutant and amiR plants were used. These articles were not cited even.

Reply 1: Thank you very much for your reasonable suggestion, there is no doubt that we agree with your comments, in order to fully understand the regulation and function of miR408 in the process of plant growth, development and reproduction, not only the evidence of overexpression of miR408 should be provided, the experimental evidence of knockout or silencing of miR408 is also very important. In fact, the complementary experiments of miR408 knockout or silencing also necessary. Thank you for recommending and supplementing the 3 studies about miR408 Knockout or silencing for us. We have cited these references in the main text (in lines 65-69, 432-458), we also have added these references in the reference list. We acknowledge your expert comments and constructive suggestions, which are valuable for improving the quality of our manuscript. 

It is well known that our understanding of plant miRNA function is still limited compared to the functions of genes encoding proteins. In addition, compared with in animals, the functions of miRNAs in plants are relatively limited, miRNAs in plants function mainly by negatively regulating target genes. One miRNA regulates multiple target genes, a target gene may also be regulated by multiple miRNAs simultaneously, it is not a one-to-one correspondence between miRNAs and target genes but a complex regulatory network. Compared to overexpression, knockout or silencing of miRNAs often cause more serious consequences including death or infertility, which makes follow-up research more difficult. Therefore, miRNA knockout or silencing test reports relatively small compared to miRNA overexpression. Overexpression of miRNA can make the negative regulation of the target gene more obvious and get a more intuitive and significant result. Thus, the existing researches on plant miRNA functions mainly focused on the regulating roles of miRNA on certain key target genes. Researches tried to reveal the signaling path between miRNA, target gene and phenotype, rather than skip target gene and directly study the relationships between miRNA and phenotype. If the studies only focused on the phenotypes and miRNAs, there will be a large gap between them, it is hard to finish a complete story. So, we have reviewed and summarized the functions of downstream target genes of miR408 in our manuscript.

Q2: Additionally, in my opinion, the authors should improve their discussion about the evolution of miR408 in the plant kingdom. The review mainly discusses the agricultural aspects of miR408 OE and more suits for submission to plant/agriculture journal (even in such case, the article's name should be changed because the article does not review miR408 roles but of its targets). Consequently, this review should be significantly improved.

Reply 2: Thank you very much for your careful review. According to your opinion, we have discussed more in-depth discussions in the evolution of miR408 in plants, and we have moved the supplemental material Fig. S1 and Fig. S2 to the main text, which makes the evolution result of miR408 more intuitive. Plant molecular biology is an important part of molecular biology. Researches on the function of miR408 in plants are limited and it is still a relatively new research field. At present, there is no systematic review of the functions of plant miR408. That’s why we think our manuscript fits the scope of IJMS. We agree with your opinion that this manuscript is suitable for possible publication in plant or agricultural journals too. However, considering the amazing high speed of review and publication and the wide range of international influence of the journal, we prefer to choose IJMS rather than other journals.

Just like what we have responded in question one (Q1), compared with in animals, the functions of miRNAs in plants are relatively limited, miRNAs in plants function mainly by negatively regulating target genes. One miRNA regulates multiple target genes, a target gene may also be regulated by multiple miRNAs simultaneously, it is not a one-to-one correspondence between miRNAs and target genes but a complex regulatory network. People try to reveal the signaling path between miRNA, target gene and phenotype, rather than skip target gene directly study the relationships between miRNA and phenotype. If the studies are only focused on the phenotypes of miRNA overexpression or knockout, there is still a large gap between the miR408 and phenotype, it is not a complete story and no regulating pathway will be discovered. This is also why we have reviewed and summarized the reasons for the miR408 downstream target gene in a review of the miR408 function. At the same time, we appreciate your suggestion, we have revised “the roles of miR408” to “the regulating roles of miR408” to make the title and content much more matching.

Minor points

Q1: miR OE does not show miR role but of its targets. 

Reply 1: Just like what we have responded to major points, different form in animals, the major role of miRNA in plants is negative regulation on the expression of target genes. Researches tried to reveal the signaling path between miRNA, target gene and phenotype, rather than skip target gene and directly study the relationships between miRNA and phenotype. If the studies only focused on the phenotypes and miRNAs, there will be a large gap between them, it is hard to finish a complete story.

Q2: Additional studies that reported the roles of miR408: Front Plant Sci. 2017; 8: 1598. doi: 10.3389/fpls.2017.01598, Plant Cell. 2021 Jul 2;33(5):1506-1529. doi: 10.1093/plcell/koab060 (STTM silencing) 

Reply 2: Thank you for recommending and supplementing these studies about miR408 Knockout or silencing for us. We have increased these contents to the main text (in lines 65-66, 68-69 and 432-444), we also have added these references in the reference list.

Q3: Lines 40-42: In my opinion, it is good to give the stage for plant researchers also. It is good to mention the works of D. Baulcombe, for example. 

Reply 3: Thank you for recommending the literatures of D. Baulcombe, et al. We have cited this content in the main text (lines 40-44), we have also added this reference in the reference list.

Q4: Lines 43-45: You are writing about conserved miRs. It is good to mention moss miRs as the sample for conserved miRs: works of Axtell or W. Frank. Better to cite original research and not reviews in this case. 

Reply 4: Thank you for recommending the literature of Axtell and W. Frank. We have cited this content in the main text (lines 46-52), we have also added this reference in the reference list.

Q5: Line 47: miR OE shows the role of its targets 

Reply 5: Thank you very much for your reasonable suggestion. In the revised manuscript we have added ‘by down-regulating the target genes’ in line of 61-63. Just like what we have responded to major points, different form in animals, the major role of miRNA in plants is negative regulation on the expression of target genes.

Q6: Line 70: ref 9,17-it is not enough to cite only these two papers if you talk about “30 species”. 

Reply 6: Thank you for your suggestion, we are sorry for our inappropriate use of words. In the revised manuscript, we have corrected the ‘reported’ into ‘annotated’ in line of 84.

Q7: Lines 75-78: I think it is a place to say something about the difference in the number of precursor and mature miRNA sequences.  Is some precursor encodes for several mature sequences? 

Reply 7: Thank you for your nice suggestion. In the revised manuscript, we have added contents about the difference in the number of precursor and mature miRNA sequences in line of 74-75.

Q8: Lines 80-90: Does miR408 exist in the non-plant kingdom? Does every species have 
the same number of miR408 (please discuss the various size of miR family in various 
species)? 

Reply 8: Thanks for your valuable comments. According to the searching results from the database, we have not got results about miR408 in the non-plant kingdom. We also failed to find relevant reports on miR408 in the non-plant kingdom. Normally, the number of miR408 (expression level) is different in different species, even in the same species, the expression of miR408 is changeable under different conditions. In addition, the size and sequence of miR408 is not completely the same in different species. About this question, we have discussed the various size of miR family in various species in line of 90-93 and 95-99.

Q9: Does miR408 exist in algae? If yes, please add it to your analysis and discuss it. 

Reply 9: Thanks for your professional suggestion. According to our evolutionary analysis of miR408, it is apparent that the miR408 family is an ancient and wildly distributed miRNAs family and bryophytes might be the evolutionary ancestors of the miR408 family, and has been strongly conserved hence forth. In the process of plant evolution, algae appeared earlier than bryophytes. In addition, we failed to find reports on miR408 in algae.

Q10: Lines 95-102: Does it important to talk about precursors? Is it important that they are different (At the end, it is mature miRNA that is very similar to others)? 

Reply 10: Good question. In our opinion, it is very important and necessary to talk about the precursors of miRNA, it determines the function of mature miRNAs. We have discussed this in line 92-95.

Q11: Lines 112-117: The number of miRs in comparison to P. patens, the evolution of the miR408 family in monocots and dicots?

Reply 11: Thank you very much for your reasonable suggestion. We have discussed it in line 139-145.

Q12: Fig. 1: check words in the fig. for ex: “trnsfer” to transfer, etc. Is it really a blue color?

Reply 12: We are sorry for this careless error. In the revised manuscript, we have corrected the ‘trnsfer’ into ‘transfer’ and corrected the ‘Blue color’ into ‘Green color’ in line of 179 and 180.

Q13: Lines 132-133: please mention that it is a prediction. I do not know any miR that 
someone showed and validated such an amount of targets. 

Reply 13: Thank you for your suggestion, we are sorry for our inappropriate use of words. Depending on your suggestion, we have corrected the ‘MiRNAs have the potential to target hundreds of mRNAs due to the imperfect complementarity needed for binding. [33]. Indeed, RNA-sequencing for miRNA targets has identified hundreds of targets for a single miRNA [19]’ into ‘MiRNAs have the potential to target hundreds of mRNAs due to the imperfect complementarity needed for binding, but only a small fraction of these interactions has been validated experimentally [33]’ in line of 182-184.

Q14: Line 200: ref 5 shows it in two places opposite to ref 21. 

Reply 14: We apologize for the careless mistake caused by the misunderstanding of this section. We have made modifications in the text of the literature (line of 250-252).

Q15: Then you are talking about OE- miR OE does not show miR role but of its targets. Recently, a nice paper was published in Plos genetics that explained it very clear, but of course, other people wrote it years ago (for ex. the works of T. Arazi). 

Reply 15: Thanks for your valuable comment. We have added these contents in the text part and have been summarized and analyzed (line of 64-65), and we have also completed in the list of references.

Q16: Lines 284-285: references about stomata and ROS. 

Reply 16: Thanks for your professional suggestion. We have made modifications in the text of the literature (line of 331-334).

Q17: Table 1: please add column OE/DR overexpression and down-regulation, silencing, knockout.  Table 1 depicts the roles of miR408 in biotic and abiotic stresses in plants, the up-regulation or down-regulation of miR408 under biotic and abiotic stress. Red represent decreased miR408 abundance under stresses, Green represent increased miR408 abundance under stresses (line of 551-552)

Reply 17:More and more evidence indicate that miRNA plays an important role in plant stress response as an important posttranscriptional regulatory factor. Through miRNA high throughput sequencing and degradome analysis, researchers have identified many miRNA and target genes related to biological and abiotic stresses in plants. More and more evidence show that miRNA is an important regulator between stress and target genes. The up-regulation or down-regulation of miR408 in Table 1 is not achieved by OE or DR, but the changes of miR408 expression under different stress conditions. Most of the research results in the table are from high-throughput sequencing, not gene overexpression or knockout results. Therefore, we do not think it is necessary to add column OE / DR in Tale 1.

Q18: Stresses: under stress conditions, many genes/proteins/miRs are differently affected in plants. It can be a secondary/third effect, not necessary that change in expression of some miR/gene contributes to resistance or sensitivity (it should be validated in mutant plants). 

Reply 18: Thank you for your professional question. With the in-depth understanding of posttranscriptional regulatory mechanisms, the functions of many regulatory factors including miRNA, lncRNA and cycle RNA have been revealed gradually. It is found that the expression of many genes or enzymes is not directly affected by the external environment, there are complex regulatory networks in these progresses. More and more evidence indicate that miRNA plays an important role in plant stress response as an important posttranscriptional regulatory factor. Through miRNA high throughput sequencing and degradome analysis, researchers have identified many miRNA and target genes related to biological and abiotic stresses in plants. More and more evidence show that miRNA is an important regulator between stress and target genes. Of course, these conclusions obtained through bioinformatics analysis need to be confirmed by molecular biology experiments. By overexpression or silencing miRNA genes and their target genes, scholars have made some progress. The main purpose of this manuscript is to summarize these latest valuable research progress.

Q19: Line 459: salt stress is complex stress, and includes several primary and secondary stresses, and affects plant at all levels, so to mention only water uptake is not proper. Especially that salt stress affects the roles of miR408 targets. 

Reply 19: Thanks for your professional suggestion, we agree with your opinion. We have corrected the ‘An excessive salt concentration in the soil affects the plants growth and development by limiting its water uptake capacity, resulting in decreases more than 20% crop production of all the world [56]’ into ‘Salinity stress is one of the major environmental stresses limiting plant growth and productivity [70] in line of 518-519.

Q20: Line 478: better antioxidant system? Is it a car? Other, more effective, etc. 

Reply 20: Thank you for your suggestion, we are sorry for our inappropriate use of words. We have corrected the ‘antioxidant system’ into ‘antioxidant capacity’ in line of 531. We also have revised the similar problems throughout the manuscript.

Q21: Line 488: these enzymes are not the targets of miR408; it can be secondary/third effects. 

Reply 21: Thank you for your suggestion, we have modified this expression in line of 527-532.

Q22: Line 492-494: too speculative. 

Reply 22: Thank you for your suggestion. we have revised this sentence in line 637-540.

Q23: Line 577: expression to expressed. Between?

Reply 23: Thank you for you patiently pointed out our language errors in the manuscript. We have corrected the ‘miR408 is differentially expression between drought-sensitive and -tolerant indica rice varieties’ into ‘different cultivars also show differential expression of miR408’ in line of 601-603.

Reviewer 2 Report

The manuscript by Gao et al. entitled “Overview of the evolution and roles of miR408 in plants” aims at reviewing the evolution of miR408 in plants and its role in regulating plants growth, development, and response to abiotic and biotic stresses. Although I find the subject of the manuscript timely and interesting for a wide range of readers, considerable improvement has to be made prior to eventual publication. The English writing has to be improved (preferably by a native speaker), and several portions of the text are repetitive (e.g., lines 1, 43 and 54; 14 and 55; 16 and 58; lines 101 and 116), redundant (e.g. lines 70 and 81; lines 101 and 102, 248 and 264), or difficult (if not impossible) to understand (e.g., lines 96-98, 111-112, 248-249; 260-263; 299-301). Moreover, the style chosen to present the information gathered is not engaging, relying too much on observations such as: “AuthorX reported that” or “AuthorY found that”. In fact, the word “reported” appears 62 times throughout the manuscript. Instead of being only a repository of data observed by other authors, this review article should compare, integrate, and discuss data collected among relevant previous studies, and advance the current knowledge on the role of miR408 in plants. I also find that the knowledge gaps identified in the Conclusion (lines 756-831) should be moved to the Discussion section. It does not make sense to present to the reader new data and bibliographic references at this stage of the manuscript.

Other comments:

  1. Provide full description of abbreviations at first mention (e.g. lines 152-156).
  2. In most cases, there is a space missing in the scientific name of plant species (e.g. lines 170, 178, 267, 318).
  3. Why is there a mention on Fig.4a in the legend of Figure 3?
  4. Genes and plant species should appear in italics (e.g. lines 571, 588, 723)
  5. Concerning Figure 4, instead of proposing a figure adapted from previously published articles, provide an original schematic representation of the role of miR408 on plant mineral metabolism to support the novelty of your manuscript. Also, why did the authors focus the role of miR408 on Cu metabolism? Why not on a poorly explored and essential micronutrient such as e.g., Fe?

Author Response

Dear editors and reviewers, 
Thank you very much for your letter with regard to our manuscript (ID: IJMS-146078). We definitely appreciate your expert comments and constructive suggestions, we have revised the manuscript point by point according to your comments, all specific items have been answered or addressed and the revisions in the text have been highlighted with red color. In addition, we have used editing services from Elsevier to decrease language mistake and improve the quality of the manuscript. In the revised manuscript, we have provided more comprehensive analysis of the present progresses and developed a more detailed discussion. I hope that our revision will enable the manuscript to meet the accepted standards of IJMS this time.

If you have any question about the revised manuscript, please don’t hesitate to contact us as soon as possible. Once again, we acknowledge your expert comments and constructive suggestions, which are valuable for improving the quality of our manuscript. 
Best Regards. 
Sincerely yours, 
Jie Xiong 

Reviewer 2

Major point:

Q1: The manuscript by Gao et al. entitled “Overview of the evolution and roles of miR408 in plants” aims at reviewing the evolution of miR408 in plants and its role in regulating plants growth, development, and response to abiotic and biotic stresses. Although I find the subject of the manuscript timely and interesting for a wide range of readers, considerable improvement has to be made prior to eventual publication.

Reply 1: Thank you very much for your positive comments on our manuscript. We have revised the manuscript point by point according to your comments, the revisions in the text have been highlighted with red color. In addition, we have used editing services from Elsevier to eliminate language mistakes and improve the quality of the manuscript. In the revised manuscript, we have provided more comprehensive analysis of the present progresses and developed a more detailed discussion.

Q2: The English writing has to be improved (preferably by a native speaker), and several portions of the text are repetitive (e.g., lines 1, 43 and 54; 14 and 55; 16 and 58; lines 101 and 116), redundant (e.g. lines 70 and 81; lines 101 and 102, 248 and 264), or difficult (if not impossible) to understand (e.g., lines 96-98, 111-112, 248-249; 260-263; 299-301).

Reply 2: We apologize for our limited English. We have rewritten the repetitive text with short and simple sentences. We also have checked the manuscript word by word to reduce typos and other language mistakes. In addition, we have used editing services from Elsevier to decrease language mistakes and improve the quality of the manuscript. Once again, thank you for your careful and patient review.

Q3: Moreover, the style chosen to present the information gathered is not engaging, relying too much on observations such as: “AuthorX reported that” or “AuthorY found that”. In fact, the word “reported” appears 62 times throughout the manuscript. Instead of being only a repository of data observed by other authors, this review article should compare, integrate, and discuss data collected among relevant previous studies, and advance the current knowledge on the role of miR408 in plants.

Reply 3: Thanks for your professional suggestion which are valuable for improving the quality of our manuscript. In the revised manuscript, we have compared, integrated, and discussed the data collected among relevant previous studies. Following your suggestion, we have summarized and discussed the current knowledge on the regulating roles of miR408 in plants. All the revisions have been highlighted with red color.

Q4: I also find that the knowledge gaps identified in the Conclusion (lines 756-831) should be moved to the Discussion section. It does not make sense to present to the reader new data and bibliographic references at this stage of the manuscript.

Reply 4: We agree with your opinion that the knowledge gaps identified in the conclusion (lines 756-831) should be moved to the discussion section. We have revised the conclusion section according to your suggestion.

Minor points:

Q1:Provide full description of abbreviations at first mention (e.g. lines 152-156).

Reply 1: Thanks very much for your valuable suggestion, we have provided full description of abbreviations at first mention in the manuscript.

Q2:In most cases, there is a space missing in the scientific name of plant species (e.g. lines 170, 178, 267, 318).

Reply 2: Thanks for your valuable suggestion. We have added a missing space in the scientific name of the plant in the manuscript.

Q3:Why is there a mention on Fig.4a in the legend of Figure 3?

Reply 3: We apologize for this careless error. We have corrected it in line 378.

Q4:Genes and plant species should appear in italics (e.g. lines 571, 588, 723)

Reply 4: Thanks for your professional suggestion, we have revised all the gene and plant species names in italics throughout the manuscript.

Q5:Concerning Figure 4, instead of proposing a figure adapted from previously published articles, provide an original schematic representation of the role of miR408 on plant mineral metabolism to support the novelty of your manuscript. Also, why did the authors focus the role of miR408 on Cu metabolism? Why not on a poorly explored and essential micronutrient such as e.g., Fe?

Reply 5: Thank you for your careful review and constructive comments. In fact, we think that the previously published figure clearly expounds the mechanism of miR408 in Cu metabolism. We just try to supplement and improve the position of LAC13 in the signaling network on this figure. According to your review comments, we found that there is still no convincing evidence for the position of LAC13 in Fig. 4, so we decided to delete Fig. 4. In addition,we are very grateful for your suggestions on miR408 regulating iron metabolism. The following detailed schematic diagram has been provided in the latest article, outlining the ways in which miR408 regulates copper and iron metabolism. We added relevant contents and cited this paper in the revised manuscript. Once again, we acknowledge your expert comments and constructive suggestions, which are valuable for improving the quality of our manuscript. 

Round 2

Reviewer 1 Report

Thank you for your changes, but I am sorry, I put this point again:

OE under constitutive promoter does not show the role of miR but of its targets. During OE (especially under constitutive promoter), miR expresses in not the right places, wrong timing, and huge amounts that usually do not exist in plants. Consequently, miR OE cannot reveal the actual role of miR in various processes, and miR OE results in plenty of artifacts.

In light of this, I will accept this article only if this point is clear. It is not your or my fault that people do what is easy. If knockout or silencing of miRNAs cause death or infertility, they can use a trans-activation system: in such way, DCL1 mutant was studied in tomato. STTM plants are available for 10 years at least, and the trans-activation system even more. Weigel lab published in 2011 sttm lines of many Arabidopsis miRs.

For ex. Paragraph from line 57: role of miR, please do not begin it from OE. Start it from loss of function, then continue to OE and mention that it is target functions.

“Regulating roles”-too wordy

I liked the addition of text in lines 432-458-it added data about miR408 functions. Moreover, anthocyanin is a very nice example that only LoF miR mutant can show the role of miR. OE of LAC3 and LoF miR408 show the same phenotype.

  1. Please consider to change the name of the article, for ex.: The functional roles of miR408 and its targets in plants

  1. Line 26- “of the evolution and regulating roles of miR408 in plants”-please add “of the evolution and regulating roles of miR408 and its targets in plants”

  1. Paragraph: Line 293: “Regulating roles of miR408 in Plant Development”

         Fig 4. The roles of miR408-regulated target genes in plant growth  and development.

You included miR408 silencing/knockout in this paragraph. Please consider to change the name of P. to: for ex.: roles of miR408  and its targets in Plant Development

This comment is relevant for all other article parts also.

  1. Lines: 362-8: It is not miR408 function because you wrote about OE or DR of UCL8, miR function is observed in loss-of-function mutants. It is about UCL8

For ex. miR160 OE and DR in tomato show opposite phenotypes-it is nice to work with such mutants. However, miR171 OE and DR show unconnected phenotypes in Arabidopsis and tomato. So, we can conclude about the miR function only from knockout or DR mutants.

  1. Lines 384-385-unclear
  2. 394-395-this is miR408 target role- and it is continued till the end of the article (for ex. lines 401-402-target function, 414-18 and etc etc…..). Please mention that it is target functions.

Minor: miR genes in italic, mature miR in roman text, for ex.: lines 384-385, is it not about mature miR? Please check it in your text. Monocots and dicots-within the sentence in lowercase (l. 139-140). Plants, green-the same-l. 180.

I am sorry for my previous comment about table 1. My fault.

Best regards.

Author Response

Dear editors and reviewers, 

Thanks for the review of our manuscript “Overview of the evolution and regulating roles of miR408 in plants (IJMS-1467087)”. We definitely appreciate your expert comments and constructive suggestions, we have revised the manuscript point by point according to your comments, all specific items have been answered and the revisions in the text have been highlighted with red color. With the help of a native English speaker, we have intensively and completely revised the language problems in the manuscript. With the help of reviewer 1, we have changed the title of the manuscript to " The evolution and functional roles of miR408 and its targets in plants ". In addition to reports on OE mutants, we have also added some reports on the function loss mutants of miR408, and discussed the importance of function loss mutants in revealing the function of miR408 in plants. I hope that our revision will enable the manuscript to meet the accepted standards this time.

If you have any question about the revised manuscript, please don’t hesitate to contact us as soon as possible. Once again, we acknowledge your expert comments and constructive suggestions, which are valuable for improving the quality of our manuscript. 
Best Regards. 
Sincerely yours, 
Jie Xiong 

Reviewer 1

Major point:

Q1: Thank you for your changes, but I am sorry, I put this point again: OE under constitutive promoter does not show the role of miR but of its targets. During OE (especially under constitutive promoter), miR expresses in not the right places, wrong timing, and huge amounts that usually do not exist in plants. Consequently, miR OE cannot reveal the actual role of miR in various processes, and miR OE results in plenty of artifacts.

Reply 1: Thank you for your professional opinions, we very much agree with your opinions. Based on your constructive comments, we have made the following revisions to the manuscript: (1) In order to distinguish the functions of miR408 and its target genes, we revised the title of the manuscript to "The evolution and functional roles of miR408 and its targets in plants". (2) In view of the fact that only using miR408 OE mutants to carry out research may cause wrong conclusions, we have added research reports related to miR408 knockout or silencing mutants in the manuscript, and analyzed and discussed the research results of these reports. (3) We believe that your comment on the wrong conclusions that may be caused by using only the miR408 OE mutant is very instructive, so this comment of yours has been added to our manuscript, and I hope you can agree.

Q2: In light of this, I will accept this article only if this point is clear. It is not your or my fault that people do what is easy. If knockout or silencing of miRNAs cause death or infertility, they can use a transactivation system: in such way, DCL1 mutant was studied in tomato. STTM plants are available for 10 years at least, and the transactivation system even more. Weigel lab published in 2011 sttm lines of many Arabidopsis miRs. For ex. Paragraph from line 57: role of miR, please do not begin it from OE. Start it from loss of function, then continue to OE and mention that it is target functions.

Reply 2: Thank you for your professional question. From your question 1, we have learned that only using miR408 overexpression mutants to study the function of miR408 is not enough and may lead to wrong research conclusions. Therefore, we must also add relevant research reports on miR408 knockout or silent mutants. Therefore, we made the following modifications: (1) We added research reports on miR408 knockout or silencing mutants; (2) We adjusted the order of miR408 function research results, start it from loss of function, then continue to OE and mention that it is target functions.

Q3: Regulating roles”-too wordy

Reply 3: We have removed “regulating” throughout this article.

Q4: I liked the addition of text in lines 432-458-it added data about miR408 functions. Moreover, anthocyanin is a very nice example that only LoF miR mutant can show the role of miR. OE of LAC3 and LoF miR408 show the same phenotype.

Reply 4: Thank you very much for your recognition and like of this part of our manuscript. According to your constructive suggestion, we have added research reports related to miR408 knockout or silencing mutants in the manuscript.

Minor points

Q1: Please consider to change the name of the article, for ex.: The functional roles of miR408 and its targets in plants

Reply 1: Thank you very much for your reasonable suggestion, we have changed the title of the manuscript to "The evolution and functional roles of miR408 and its targets in plants".

Q2: Line 26- “of the evolution and regulating roles of miR408 in plants”-please add “of the evolution and regulating roles of miR408 and its targets in plants”

Reply 2: Thanks for your professional suggestion. We have changed the “of the evolution and regulating roles of miR408 in plants” into “of the evolutionary and regulatory roles of miR408 and its targets in plants”.

Q3: Paragraph: Line 293: “Regulating roles of miR408 in Plant Development” Fig 4. The roles of miR408-regulated target genes in plant growth and development. You included miR408 silencing/knockout in this paragraph. Please consider to change the name of P. to: for ex.: roles of miR408 and its targets in Plant Development. This comment is relevant for all also.

Reply 3: Thank you for your suggestion, we have changed the subtitle and figure legend according to your suggestion.

Q4: Lines: 362-8: It is not miR408 function because you wrote about OE or DR of UCL8, miR function is observed in loss-of-function mutants. It is about UCL8. For ex. miR160 OE and DR in tomato show opposite phenotypes-it is nice to work with such mutants. However, miR171 OE and DR show unconnected phenotypes in Arabidopsis and tomato. So, we can conclude about the miR function only from knockout or DR mutants.

Reply 4: Thank you for your comments, we have learned that only using miR408 OE mutants to study the function of miR408 is not enough and may lead to wrong research conclusions, we can conclude about the miR function only from knockout or DR mutants. We added “roles of miR408 and its targets in plants” to emphasize the functions of targets in the revised manuscript, we also revised the sentences from 362 to 368.

Q5: Lines 384-385-unclear

Reply 5: We apologize for the unclear language in our manuscript. We have changed the “MiR408 plays crucial roles in regulating seed size in plants. In A. thaliana and N. tabacum transgenic lines overexpression of miR408, the seed size was significantly increased compared with that of WT.” into “In A. thaliana and N. tabacum, the overexpression of miR408 significantly increased seed size and seed weight compared with the WT [19, 21].”

Q6: 394-395-this is miR408 target role- and it is continued till the end of the article (for ex. lines 401-402-target function, 414-18 and etc etc…..). Please mention that it is target functions. Minor: miR genes in italic, mature miR in roman text, for ex.: lines 384-385, is it not about mature miR? Please check it in your text. Monocots and dicots-within the sentence in lowercase (l. 139-140). Plants, green-the same-l. 180.

Reply 6: Thank you for your suggestion, we have mentioned that it is target functions. We have revised miR genes in italics throughout the manuscript. We also corrected “Dicots, Monocots, Plants and Green” into “dicots, monocots and green”.

Reviewer 2 Report

The manuscript by Gao shows considerable improvement, but I recommend further efforts prior to eventual publication. I found that the English writing was not sufficiently improved, and I tried to annotate as much changes as possible in the attached file, but I strongly suggest further revision from an English native speaker. Some portions of the text are written in a careless manner, which unfortunately decreases the overall quality of the work and diminishes it scientific soundness. For example, some portions of the text are redundant (lines 382, 384, 390), and it frequently feels like a repository of information published by other authors (e.g lines 491, 492: “Sunkar and Zhu [10] first reported that…. Ma et al. firstly reported that…”). Additional efforts should be employed to critically discuss the results. Also, please provide the full description of abbreviations at first mention (e.g. GA, ROS, SOD, CAT, POD, ABA, GA3, Salb, RA, Fv/Fm, WT) and provide the full name names of plant species the first time they are refereed (e.g. S. miltiorrhiza, line 445). Format all gene names in italics (e.g. lines 338, 434, 437, 522), but not enzymes (e.g. line 530). References seem to be missing from lines 384-389 and 391-393. In the section related to the role of miR408 on nutrient deficiency, a great deal of attention has been paid to copper, which is not even an essential nutrient, whereas other minerals important for plant nutrition, such as zinc, were overlooked (e.g. the work from Li et al. New Phytologist (2013) 200: 1102–1115, working with microRNAs in Sorghum bicolor subjected to zinc deficiency was not included, just as an example). In my view, the manuscript has great potential for publication, but its writing style should be improved, and its content better supported with additional bibliography before publication.

Author Response

Dear editors and reviewers, 

Thanks for the review of our manuscript “Overview of the evolution and regulating roles of miR408 in plants (IJMS-1467087)”. We definitely appreciate your expert comments and constructive suggestions, we have revised the manuscript point by point according to your comments, all specific items have been answered and the revisions in the text have been highlighted with red color. With the help of a native English speaker, we have intensively and completely revised the language problems in the manuscript. With the help of reviewer 1, we have changed the title of the manuscript to " The evolution and functional roles of miR408 and its targets in plants ". In addition to reports on OE mutants, we have also added some reports on the function loss mutants of miR408, and discussed the importance of function loss mutants in revealing the function of miR408 in plants. I hope that our revision will enable the manuscript to meet the accepted standards this time.

If you have any question about the revised manuscript, please don’t hesitate to contact us as soon as possible. Once again, we acknowledge your expert comments and constructive suggestions, which are valuable for improving the quality of our manuscript. 
Best Regards. 
Sincerely yours, 
Jie Xiong 

Reviewer 2

Q1: The manuscript by Gao shows considerable improvement, but I recommend further efforts prior to eventual publication. I found that the English writing was not sufficiently improved, and I tried to annotate as much changes as possible in the attached file, but I strongly suggest further revision from an English native speaker.

Reply 1: Thank you very much for your careful comments. We have revised the manuscript point by point according to your comments. With the help of a native English speaker, we have intensively and completely revised the language problems in the manuscript. 

Q2: Some portions of the text are written in a careless manner, which unfortunately decreases the overall quality of the work and diminishes it scientific soundness. For example, some portions of the text are redundant (lines 382, 384, 390),

Reply 2: Thank you very much for your reasonable suggestion. We are sorry that there is a text content redundancy in our text, in order to make the manuscript more concise and clear we have modified these redundant parts in the revised version.

Q3:and it frequently feels like a repository of information published by other authors (e.g lines 491, 492: “Sunkar and Zhu [10] first reported that…. Ma et al. firstly reported that…”). Additional efforts should be employed to critically discuss the results.  

Reply 3: Thank you very much for your reasonable suggestion. We have rewritten those text contents. In addition, in the revised manuscript, we have developed a more rigorous structure for the discussion, the revised manuscript is no more a simple summary of literature, detailed hypothesized mechanisms, conclusive outcomes and critical evaluations have been supplied.

Q4: Also, please provide the full description of abbreviations at first mention (e.g. GA, ROS, SOD, CAT, POD, ABA, GA3, Salb, RA, Fv/Fm, WT) and provide the full name names of plant species the first time they are refereed (e.g. S. miltiorrhiza, line 445). Format all gene names in italics (e.g. lines 338, 434, 437, 522), but not enzymes (e.g. line 530). References seem to be missing from lines 384-389 and 391-393.

Reply 4: Thank you very much for your reasonable suggestion. We apologize for those careless error. We have provided full description of abbreviations at first mention in the text. We also have revised all the gene and plant species names in italics throughout the manuscript.

Q5: In the section related to the role of miR408 on nutrient deficiency, a great deal of attention has been paid to copper, which is not even an essential nutrient, whereas other minerals important for plant nutrition, such as zinc, were overlooked (e.g. the work from Li et al. New Phytologist (2013) 200: 1102–1115, working with microRNAs in Sorghum bicolor subjected to zinc deficiency was not included, just as an example).

Reply 5: MiR408 interacts with genes encoding Cu-containing proteins such as cupredoxin, PLC, UCL and LAC, all of which belong to the phytocyanin family. In addition, an increasing number of studies have shown that miR408 is responsive to Cu deficiency in plants. It’s inferred that miR408 is the main regulator in Cu homeostasis. Many of the target genes regulated by miR408 belong to Phytocyanin (PCs). PCs are blue Cu proteins that bind Cu ion. It is not only related to the activity of the electron carrier, but also has a great influence on the growth and stress resistance of plants. For these results, many reports focused on Cu in investigating the role of miR408. Thank you for recommending the literatures of Li et al. We added this content in the main text and the reference list. In addition, we also added some articles related to the role of miR408 on other minerals such as iron in the main text.

Q6: In my view, the manuscript has great potential for publication, but its writing style should be improved, and its content better supported with additional bibliography before publication.

Reply 6: Thank you for your nice suggestion. With the help of a native English speaker, we have systematically revised the writing of the manuscript to make the manuscript more concise, clear and logical. In addition, we have added research reports related to miR408 knockout or silencing mutants in the manuscript, and analyzed and discussed the research results of these reports. We adjusted the order of miR408 function research results, start it from loss of function, then continue to OE and mention that it is target functions.
